# Monumental rock art illustrates that humans thrived in the Arabian Desert during the Pleistocene-Holocene transition

Maria Guagnin [1] ✉, Ceri Shipton [2,3] ✉, Faisal Al-Jibreen[4], Giacomo Losi[5], Amir Kalifi [5], Simon J. Armitage [6,7], Finn Stileman[8], Mathew Stewart[9], Fahad Al-Tamimi[4], Paul S. Breeze [10], Frans van Buchem[5], Nick Drake[10], Mohammed Al-Shamry[4], Ahmed Al-Shammari[4], Jaber Al-Wadani[4], Abdullah M. Alsharekh [11] & Michael Petraglia [9,12,13]

Dated archaeological sites are absent in northern Arabia between the Last Glacial Maximum (LGM) and 10,000 years ago (ka), signifying potential population abandonment prior to the onset of the Holocene humid period. Here we present evidence that playas became established in the Nefud desert of northern Arabia between ~16 and ~13 ka, the earliest reported presence of surface water following the hyper-aridity of the LGM. These fresh water sources facilitated human expansions into arid landscapes as shown by new excavations of stratified archaeological sites dating to between 12.8 and 11.4 ka. During the Pleistocene-Holocene transition, human populations exploited a network of seasonal water bodies - marking locations and access routes with monumental rock engravings of camels, ibex, wild equids, gazelles, and aurochs. These communities made distinctive stone tool types showing ongoing connections to the late Epipalaeolithic and Pre-Pottery Neolithic populations of the Levant.

At the end of the last Ice Age, the Last Glacial Maximum (LGM, ~25–20 ka) ushered in harsher, cooler climatic conditions that resulted in extreme aridity across the Middle East, causing widespread dune mobilisation and de-population on the Arabian peninsula[1–4]. In the "Fertile Crescent", the Pleistocene-Holocene transition brought a return to more favourable climatic conditions. The Natufian (~14.6 – 11.5 ka) is associated with intensive plant exploitation, and the widespread adoption of sedentism and food storage[1,2,5]. The domestication of plants and animals commenced in the Pre-Pottery Neolithic (PPN) A

and B, respectively (~11.7–10.5 ka and 10.5–8.25 ka)[1,6–8]. These pronounced socio-economic shifts went hand in hand with the emergence of a rich symbology expressed in carvings, figurines, and architecture[1,9]. Notably, distinct geographic differences formed across the late Epipalaeolithic and the emerging Neolithic. In the Mediterranean woodland zones of the Levant, populations were mostly sedentary, while in the more arid regions of the Negev, Sinai, and eastern Jordan, populations were more mobile and reliant on hunting[5,6]. How climate and population dynamics during this crucial period in human

[1]Department of Archaeology, Max Planck Institute of Geoanthropology, Jena, Germany. [2]Institute of Archaeology, University College London, London, UK. [3]College of Asia and the Pacific, Australian National University, Canberra, ACT, Australia. [4]Heritage Commission, Ministry of Culture, Riyadh, Saudi Arabia. [5]Physical Science and Engineering Division, King Abdullah University of Science and Technology (KAUST), Thuwal, Saudi Arabia. [6]Department of Geography, Royal Holloway University of London, London, UK. [7]SFF Centre for Early Sapiens Behaviour (SapienCE), University of Bergen, Bergen, Norway. [8]Department of Archaeology, University of Cambridge, Cambridge, UK. [9]Australian Research Centre for Human Evolution, Griffith University, Brisbane, QLD, Australia. [10]Department of Geography, King's College London, London, UK. [11]Department of Archaeology, College of Tourism and Archaeology, King Saud University, Riyadh, Saudi Arabia. [12]Human Origins Program, Smithsonian Institution, Washington, DC, USA. [13]School of Social Science, University of Queensland, Brisbane, QLD, Australia. ✉e-mail: guagnin@gea.mpg.de; c.shipton@ucl.ac.uk

history unfolded further south, on the Arabian Peninsula, is poorly understood.

The presence of human populations in northern Arabia has been primarily linked with episodic humid phases[10,11]. Evidence from palaeolake deposits and associated pollen and molluscs indicate increased rainfall and vegetation during the Holocene humid period, ~10–6 ka[11–16]. Several sites document human occupation of the region between 10 and 8 ka, such as in the Umm Jirsan lava tube in the Harrat Khaybar[17], and the sites of Al Rabyah[12], Jebel Qattar[18], and Jebel Oraf[19], all located in the Jubbah oasis. Radiocarbon ages from Neolithic sites across north-western Arabia suggest a peak in human activity between 7.6 and 6.8 ka, which corresponds with the end of the Holocene humid period[19–23]. Neolithic communities built large stone structures, including large hunting traps known as desert kites, and large rectangular ritual structures called mustatils, which stretch up to 620 m in length[21,23,24]. Pastoral communities also created a rich record of rock art, with thousands of depictions of wild animals such as ibex and livestock such as cattle. These are typically stylised, with human figures being elongated with extremely thin arms, and animal depictions with exaggerated horns[25,26].

Prior to the onset of the Holocene humid period, little is known about the relatively arid period spanning the end of the Pleistocene and the earliest Holocene in Arabia. An absence of dated archaeological sites has led to a presumed absence of human occupation of the Arabian interior. However, superimpositions in the rock art record appear to show earlier phases of human activity, prior to the arrival of domesticated livestock[25].

In 2022, a rock art site was discovered at Sahout (SAU), an area south of the Nefud desert. Here, sandstone outcrops form low-lying mountain ranges that are separated by scattered sand dunes and rocky pavements (Fig. 1). Survey documented 18 life-sized engravings of camels, ibex, and equids, with stratigraphic analysis suggesting that multiple large camel engravings may predate Neolithic engravings and the Holocene humid period. However, test excavations did not permit a correlation between dated deposits and rock art[27]. Three further panels with large camels were recently documented by members of the public in localities ca. 25 km further south (Fig. 1). Their

coordinates were compiled by the Saudi Heritage Commission and shared with our team. This information provided an opportunity to document additional panels belonging to this possible early Holocene rock art tradition, and to survey and excavate any associated archaeological deposits.

Here we report the results of archaeological surveys and excavations at rock art sites with life-sized camel engravings south of the Nefud desert, as well as analyses of spatially associated playa deposits. Three archaeologically unexplored areas were visited during our fieldwork in 2023: Jebel Arnaan (ARN), Jebel Mleiha (MLH), and Jebel Misma (JMI). The three areas span ~30 km along the southern edge of the Nefud desert (Fig. 1). To acquire paleoenvironmental context for the region, playa sediments were excavated and dated at ARN and JMI (Fig. 1). The primary aim of this research was to test if the rock art panels and archaeological deposits represent earlier occupations during the Holocene humid period, resulting in a condensed chronology of rock art production, or if they belong to earlier periods and represent a longer period of human presence in northern Arabia.

## Results
### Regional geomorphology and palaeoenvironments
To evaluate the terminal Pleistocene and early Holocene environments of the region, trenches were excavated at four playas in the vicinity of the archaeological sites, two of which contained sufficient sediment for palaeoenvironmental analysis and luminescence dating: ARN (Site 1) and JMI (Site 4), (Fig. 1; Supplementary Note 5 and Supplementary Fig. 42). As local centres of deposition, and areas where water accumulated in the past, these playas provide archives of hydroclimate. Gravels found at the base of both 2 m deep trenches were poorly sorted, containing pebbles of up to 5 cm, and were interpreted as alluvial fan deposits. A luminescence age of 68.8 ± 5.0 ka (MIS-B-1) from JMI indicates that these sediments were deposited at the Marine Isotope Stage 5a to 4 transition (Table 1). Palaeohydrological activation at this time is consistent with both broader regional records[28] and a surface find of a Middle Palaeolithic Levallois core from JMI (Supplementary Fig. 32).

At both excavated paleoenvironmental sites, the overlying sediments consist of a thin interval of alternating much finer-grained and

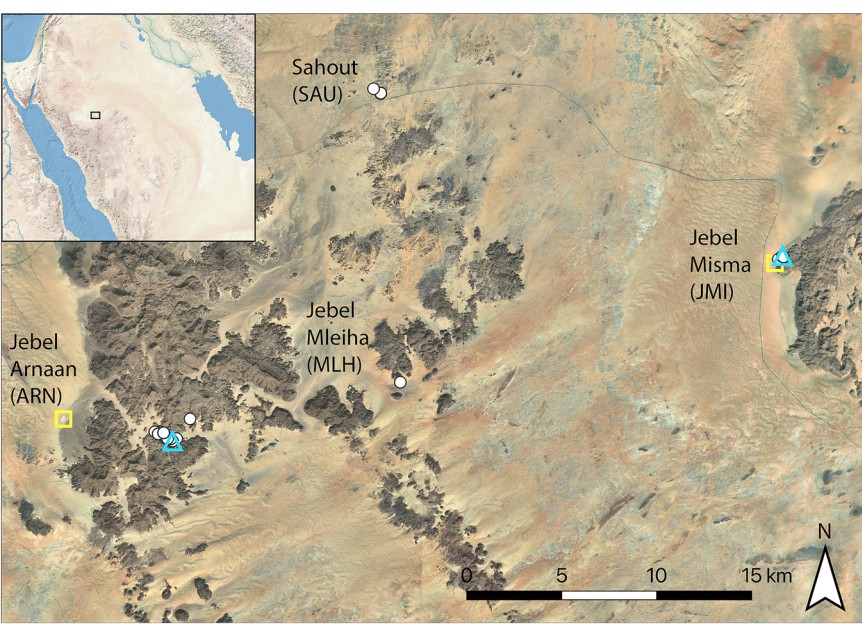

**Fig. 1 | Map of the Sahout region south of the Nefud desert, Hail Province, northern Saudi Arabia, showing the rock art areas mentioned in the text.** White dots: locations of rock art panels; ARN: 46 panels with 64 life-sized animals, 15 human figures and 22 small or partial figures; MLH: 1 panel with 1 life-sized camel, JMI: 14 panels with 48 life-sized animals (23 on a single panel – JMI18), 1 human figure and 4 small or partial figures, SAU: 18 life-sized animal engravings recorded in 2022[27], 3 human figures and 1 small figure. Blue triangles: archaeological excavations; yellow squares: playa excavations with sufficient sediment for analysis. Bing Virtual Earth imagery as basemap in QGIS. Imagery © 2025 Microsoft Corporation. Inset: Natural Earth.

**Table 1 | Luminescence ages from archaeological sites and playa deposits**

| Sample | Site | Context | Grain size (µm) | Depth (cm) | Age (ka) |
|---|---|---|---|---|---|
| MIS-B-1 | Playa, Jebel Misma (Site 4) | 185 cm below the ground surface, sand matrix in gravels below the playa deposit | 180–210 | 185 | 68.8 ± 5.0 |
| MIS-B-2 | Playa, Jebel Misma (Site 4) | 170 cm below the ground surface, sand layer between claystone of playa deposit | 180–210 | 170 | 15.5 ± 1.5 |
| MIS-B-3 | Playa, Jebel Misma (Site 4) | 140 cm below the ground surface, sand layer between claystone of playa deposit | 180–210 | 140 | 17.1 ± 1.7 |
| SAH-L3-1 | Playa, Jebel Arnaan (Site 1) | 180 cm below the ground surface, unconsolidated sand layer below playa deposit | 180–210 | 180 | 12.7 ± 2.1 |
| SAH-L3-2 | Playa, Jebel Arnaan (Site 1) | 60 cm below the ground surface, quartz-rich material in claystone playa deposit | 4-11 | 60 | 7.6 ± 1.0 |
| ARN-T1-2 | Archaeological site, Jebel Arnaan (ARN3) | 153 cm below the original surface, directly above the engraving stone; layer 5 | 180–210 | 134 | 12.2 ± 1.4 |
| ARN-T1-3 | Archaeological site, Jebel Arnaan (ARN3) | 142 cm below original surface, bottom of archaeological remains; layer 5 | 180–210 | 124 | 12.8 ± 1.1 |
| JMI8-T1-4 | Archaeological site, Jebel Misma (JMI8) | layer 6, moderate artefact density; lowest layer with a matrix that allowed sampling | 180–210 | 45 | 12.0 ± 1.8 |

well-sorted aeolian sands and playa deposits. These are followed by a prolonged sequence of playa deposits consisting of quartz sand, clays (illite and kaolinite), and calcite, indicating more humid conditions with increased water accumulation in these local depressions. We interpret the onset of playa sediment accretion as representing the change in the balance between aeolian erosion and fluvial sedimentation. During the LGM, the hyper-arid environment meant that any fine-grained sediments deposited in ephemeral floods were subsequently eroded by the wind. As the climate became less arid, fluvial sedimentation increased, becoming greater than the aeolian erosion rate and sediments started to accumulate. However, the subordinate carbonate concentration and the absence of root traces and organic-rich layers suggest that conditions remained too dry for the establishment of more permanent water bodies, indicating that a dryland environment persisted around these ephemeral, and probably seasonal, lakes. Luminescence ages constrain the onset of surface water to between 17.1 ± 1.7 ka (MIS-B-3) and 15.5 ± 1.5 ka (MIS-B-2) at JMI (Site 4) and to 12.7 ± 2.1 ka (SAH-L3-1) at ARN (Site 1) (Table 1). The onset of sedimentation after a long hiatus, coupled with an increase in calcite noted in both sedimentary records, indicates that local conditions gradually became more humid after ~16 ka at JMI to ~13 ka at ARN, with these site-specific differences in timing likely due to differences in catchment hydrology (Table 1). These sites thus represent the earliest evidence from northern Arabia of increased humidity following the hyper-arid LGM. However, the lakes were ephemeral, indicating an arid or semi-arid climate prevailed in the region.

## Monumental rock art

Surveys at ARN and JMI identified previously unknown rock art landscapes with life-sized depictions of wild mammals and human figures, and an individual panel at MLH. Across the three areas 62 rock art panels were recorded, containing 176 engravings. Of these, 130 were life-sized and naturalistic engravings depicting camels (90), ibex (17), equids (15), gazelles (7), and aurochs (1), with individual representations frequently measuring up to 2.5–3.0 m in length and 1.8–2.2 m in height. In addition, we identified 2 camel footprints, 15 smaller scale naturalistic depictions of camels, 19 human figures, 4 human faces or masks, and 6 unidentified, partial engravings (Supplementary Data 1). Most of the recorded engravings show camels in a detailed and naturalistic style that echoes the reliefs of the Camel Site to the north of the Nefud desert. This includes the frequent depiction of a bulging neckline, indicating they represent male camels in rut[29,30] (Fig. 2A, B).

The depictions span multiple engraving phases, with images often overlapping on rock surfaces. Sometimes this was done to update an existing representation (Fig. 2B) or to depict a different animal species (Fig. 2C). We distinguish four phases here. Two early rock art phases: small, stylised depictions of women (phase 1, traced in green), followed by large human figures in frontal view (phase 2, traced in yellow). These human figures were always noted to be older than, i.e. underneath, the recorded life-sized animal representations (Fig. 2A), and they make up a much smaller proportion of motifs. The third phase shows detailed, extremely naturalistic representations of animals, where each depiction has individual characteristics (traced in white) (Fig. 2A, B). A later, fourth phase (traced in blue) shows more stylised depictions of animals with cartoonish features, including rounded eyes and horn ridges, and more standardised, near-identical depictions of animals (Figs. 2C, D and 3C).

Unlike the SAU site, where many of the engravings were found inside narrow gaps between boulders[27], engravings at ARN and JMI were found on prominent locations on boulders or cliff surfaces, facing into the landscape (Supplementary Figs. 1 and 7). Some of these panels were etched onto cliff surfaces in inaccessible but highly visible locations (Fig. 3). The difficulty in getting to and engraving these rock surfaces, and their enhanced visibility by height were clearly attractive for the engravers. The precarious nature of the engraving process is particularly evident in the largest recorded panel. On the ground,

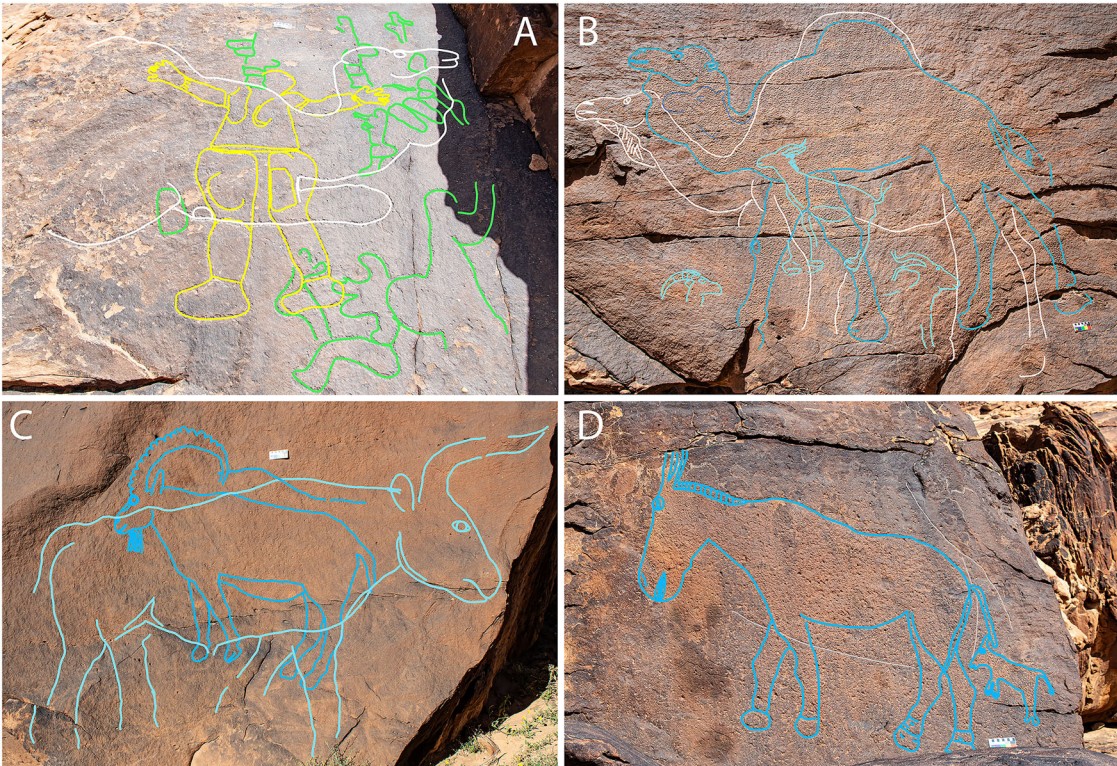

**Fig. 2 | Rock art panels at Jebel Arnaan.** Tracings highlight the layering of engravings, showing phase 1 in green, phase 2 in yellow, phase 3 in white and phase 4 in shades of blue. Rock art scale is 10 cm wide. **A** Panel ARN21A: Several small, stylised depictions of women (phase 1, traced in green), superimposed with a large human figure (phase 2, traced in yellow). Large, kneeling camel engraved over the top (phase 3, traced in white). **B** Panel ARN22A : naturalistic camel (phase 3, traced in white), superimposed by a camel with stylised, rounded eye and standardised outline (phase 4, traced in blue); original and unfinished camel head (traced in dark blue); three gazelle engravings were added during a later part of phase 4 (traced in light blue). Additional examples are provided in Supplementary Fig. 4. **C** Panel ARN3B : ibex with cartoon-like eye and horn (phase 4, traced in blue), superimposed with life-sized aurochs (phase 4, traced in lighter blue). **D** Panel ARN 39 : equid with cartoon-like eye, and with a young (phase 4, traced in blue). Untraced photos are provided in Supplementary Figs. 2 and 3.

panel JMI18 is today only visible in optimal light conditions for about 1.5 h in the morning, due to its elevated location and the varnish build-up on the engravings. This panel would have been accessed by climbing up a cliff and then engraved while standing on a downward sloping ledge, only ~30–50 cm in width (Fig. 3B). Today the sandstone is too degraded to reach the ledge safely, and the panel was documented using a drone. The friable nature of the substrate and the slope of the narrow ledges suggest the engravers likely risked their lives to create this art. Engraving at close range would have required them to use direct percussion, while also preventing them from being able to see the complete image. Twenty-three life-sized camels and equids, each with an individual length of ~1.7–2.6 m were engraved on this surface (Fig. 3C), with the engravings stretching ~23 m across two cliff surfaces at a height of 34 m and 39 m (Fig. 3A), giving this rock art a monumental scale.

All recorded rock art panels show a thick coating of dark rock varnish on the natural sandstone surface and inside the engraved lines (Fig. 2), although on some panels this varnish has partially eroded and is now only visible in areas that are less exposed (Fig. 3C and Supplementary Fig. 3C, D). The re-formation of rock varnish following the exposure of the fresh sandstone during the engraving process has been shown to take over 8 ka[31,32], and provides a first indication of the antiquity of these images.

The appearance of multiple life-sized animal engravings is impressive today. Freshly engraved against the varnish, the images would have had considerable visual impact. The durability of the images may have facilitated the remembrance of meaning and symbolism across generations of people using these sites. The large naturalistic engravings, therefore, align with the definition of monumentality, which references great size and effort, but also longevity and remembrance within a community[33,34].

## Archaeological Excavations

Four trenches revealed stratified archaeological deposits. Two were excavated at ARN (ARN3, T1 and T2), and two at JMI (JMI7 and JMI8, Fig. 1), yielding artefacts including over 1200 lithics (Supplementary Note 2), and 16 bone fragments (Supplementary Note 4). Optically stimulated luminescence (OSL) and radiocarbon dating (Supplementary Note 6) samples from the archaeological deposits attest to human occupation during the terminal Pleistocene and earliest Holocene.

In front of panel ARN3, a 2 × 1 m trench (T1) was excavated directly below two life-sized engravings of camels, one superimposed on the other (Fig. 4). Below recently disturbed sediments, the excavation revealed intact archaeological deposits with layer 5 in particular containing dozens of flaked stone artefacts. This layer was sealed by several large sandstone fragments in layer 4 (Fig. 4), likely the result of a colluvial event. In ARN3 T1, the legs of the engraved camels were buried at a depth indicating that they were engraved no later than when layer 4 was deposited (Fig. 4). Layer 5 included a stone tool that may have been used for engraving the rock art (discussed below) (Fig. 4, orange). Two luminescence samples obtained from layer 5, dating to 12.2 ± 1.4 ka, and 12.8 ± 1.1 ka (Table 1), directly date the burial of the artefacts and pecking tool, and provide an indirect date for the engraving above. Analysis of the engraved lines shows that all images were made using a pecking stone. In some engravings, pecked lines were smoothed in a second step (Fig. 4 inset).

A second trench (T2) was excavated at ARN3 1.2 m to the east, beneath two more camel engravings at the other end of the same

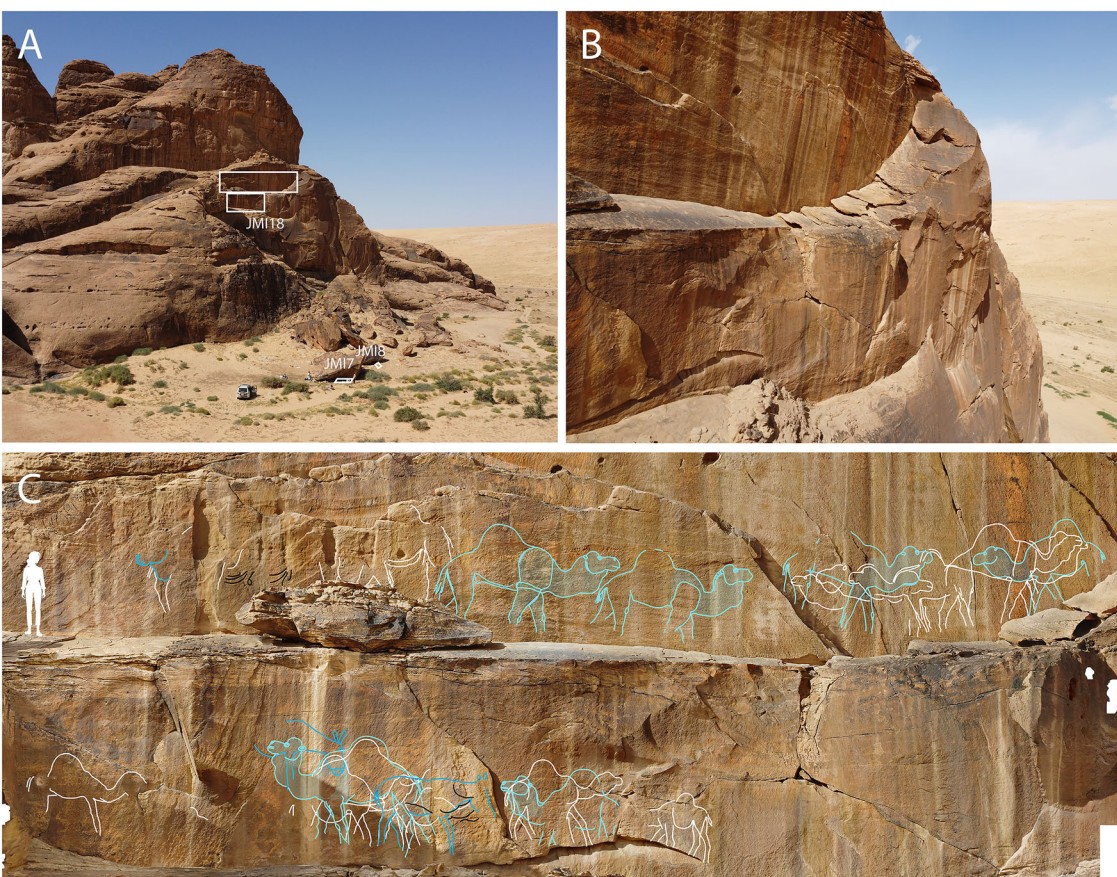

**Fig. 3 | Monumental rock art panel at Jebel Misma (JMI18). A** Location of the panels at 34 m and 39 m height, and trenches JMI7 and JMI8 (project vehicle for scale). An OSL date of 12.0 ± 1.8 was obtained from JMI8 (Table 1: JMI8-T1-4). Excavated playa deposits are located directly behind this spur (see also Figs. 1 and 8). **B** narrow, downward sloping ledges in front of the panel viewed from above. **C** orthophoto generated via a high-resolution 3D model of the panel, with tracings showing 19 life-sized camels and 3 equids (one further camel was

documented on a collapsed fragment left of the image). Naturalistic animals belonging to phase 3 traced in white. More stylised and standardised depictions of phase 4 traced in blue, including two engravings of equids traced in dark blue, and superimposed stylised camels traced in light blue. Unidentified lines traced in black. White traced camels: 1.7–1.9 m length, blue camels: 2.15–2.6 m length. A human figure was added on the far left for scale (1.7 m).

boulder surface (Supplementary Fig. 11). T2 also revealed layers containing hundreds of lithics (layers 6–8; Supplementary Fig. 13) with three small hearths in layer 7, one of which was radiocarbon dated to 11.44 ± 0.18 ka (cal. BPUGAMS65278, Table 2). Refits from layers 8 and 9 in T2 testify to the high integrity of the archaeological contexts (Supplementary Note 3).

At JMI, two trenches were excavated beneath boulders featuring rock art panels. At JMI7 a 2 × 1 m trench was excavated beneath life-sized engravings of two camels and an equid, which revealed a single layer with high artefact concentrations (layer 4; Supplementary Fig. 16). At JMI8 a 2 × 1 m trench was excavated below a life-sized engraved camel. Here moderate artefact concentrations were interspersed throughout the stratigraphic sequence, with the highest artefact concentration in layer 2, near the surface (Supplementary Fig. 19). A luminescence sample was obtained from layer 6, which provided an age of 12.0 ± 1.8 ka (JMI8-T1-4, Table 1), consistent with the luminescence and ¹⁴C ages at ARN.

### Artefact Assemblages

The ARN3 excavations revealed a rich assemblage of occupation debris. In T1, 101 lithics were recovered, with over half in layer 5 (Fig. 4), while in T2, 532 lithics were recovered, with over 95% from layers 5–9 (Supplementary Fig. 13). Chert was the main material used, constituting 36% of the pieces, with obsidian and crystal quartz making up another 23% and 15% respectively (Supplementary Data 2). Lithics

showed a diversity of reduction strategies, including small bipolar cores occurring alongside chert bladelets/blades with lamellar scar patterns and often ground platforms (Supplementary Note 3.1).

A total of 30 retouched pieces were recovered. Among these, notches were common with eight notched pieces and two opposed notch blades (Supplementary Table 1). Two chert drills were recovered in T2, as well as a single endscraper from the disturbed upper part of T1. The largest retouched artefacts were two tongue-shaped scrapers from T2 layer 8 (Supplementary Fig. 28). T2 produced seven marginally retouched convergent points made on chert bladelets, while the tanged butt of a point on an obsidian blade was recovered from T1 (Supplementary Figs. 25 and 26). The most distinctive artefacts were a single crystal quartz Helwan bladelet from T1 layer 4 (Fig. 5B), and three chert El Khiam points, a broken butt from T1 layer 5, and both a broken butt and a complete El Khiam point from T2 layer 5 (Fig. 5A and Supplementary Fig. 26). The luminescence age of 12.2 ± 1.4 ka (ARN-T1-2, Table 1) in T1 and the radiocarbon date of 11.44 ± 0.18 ka (UGAMS65278, Table 2) from T2 correspond with the age of El Khiam points from the Levant and therefore indicate that the main occupation was contemporary with the PPNA.

Excavations at JMI7 produced 100 lithics while JMI8 produced 519 (Supplementary Figs. 16 and 19). Quartz was the most commonly used material comprising 65% of pieces, with an additional 20% made from silcrete (Supplementary Data 3). The single core was a silcrete naviform piece from layer 4 in JMI7 (Supplementary Fig. 29). The JMI7 and

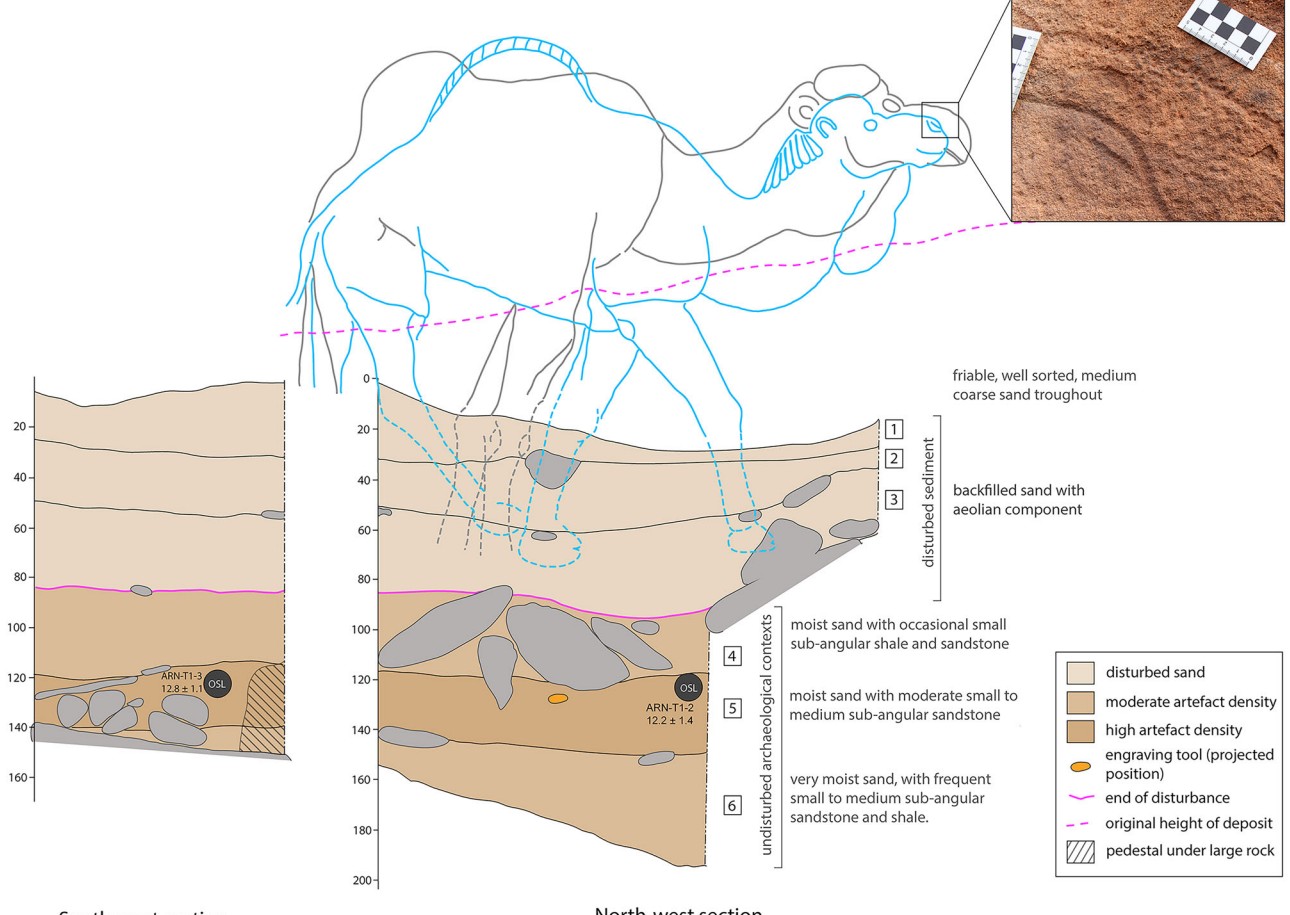

**Fig. 4 | Excavations at ARN3 Trench 1.** South-western and north-western sections, highlighting the extent of the disturbance: original height of deposit (pink dotted line), and extent of disturbed sand (pink line). Colour grading of layers reflects artefact density. Location of OSL samples shown in black, location of engraving tool projected onto section from a plotted position 72 cm south-east. Earlier camel engraving (phase 3) traced in grey, later camel engraving (phase 4) traced in blue, dashed lines indicated parts of the engraving that were covered with sediment. Note that the legs of the earlier (grey) camel have partially eroded and are no longer visible. Inset shows a closeup of the engraved lines, the crudely pecked line of the earlier (grey) camel on the upper right, and the pecked and smoothed lines of the nose of the later (blue) camel on the lower left. Remains of dark rock varnish[29,31,47] can be seen in several patches on both lines.

### Table 2 | Radiocarbon date from excavations at ARN

| Lab code (UGAMS) | Context | Material | δ¹³C,‰ | pMC | ¹⁴C age years BP | calibrated date (calBP) |
|---|---|---|---|---|---|---|
| 65278 | ARN3, T2, context 15, layer 7 | charcoal | −11.62 | 28.89 ± 0.11 | 9970 ± 30 | 11613–11526 (16.2%) 11505–11421 (19.5%) 11412 - 11264 (59.7%) |

Uncalibrated date is given in radiocarbon years before 1950 (years BP), using the 14 C half-life of 5568 years. The error is quoted as one standard deviation and reflects both statistical and experimental errors. The date has been corrected for isotope fractionation. Charcoal sample calibrated using OxCal calibration programme v4.4.4, which uses the IntCal20 calibration curve[56], and 95.4% probability. Probability for calibrated age ranges is indicated in brackets.

JMI8 excavations produced nine retouched artefacts (Supplementary Table 2). An opposed-notch chert blade was recovered in layer 5 (Supplementary Fig. 31), with two other chert notched blades recovered from JMI8 layer 4, but these did not have the distinctive opposing notches. This same layer also produced a chert drill with a distal break. A broken Helwan point was found on the surface between JMI7 and JMI8 (Supplementary Fig. 30).

Several of the recovered artefacts have associations with the PPN and even the Natufian in the Levant. Helwan bladelets (Fig. 5B) and opposed notch blades are known from Natufian sites; El Khiam points (Fig. 5A) are an artefact type characteristic of the PPNA, whereas Helwan points and naviform cores are typical for the PPNB (Supplementary Note 3). These artefact types accord with the chronometric dating at ARN and JMI (Tables 1 and 2).

Four grinding stones were recovered from the ARN3 excavations and six from the JMI excavations, along with a facetted stone ball (Supplementary Fig. 34) and two stone platters in JMI7 layer 4 (Supplementary Fig. 35). Two mortars were documented on an exposed rock surface between JMI7 and JMI8 (Supplementary Fig. 33). Mortars are typical of the PPNA[35] and have not previously been documented among grindstone assemblages of Neolithic northern Arabia[36].

A total of five ground stone beads were recovered, four green pieces from ARN T2 layers 7 and 9, and one from JMI8 layer 2. The recovery of two broken beads and one unfinished bead, together with drills at both sites, suggests on-site manufacture. Two marine dentalium (tooth-shell) beads were recovered from ARN3 T2, layers 7 and 8 (Fig. 5F and Supplementary Fig. 41). Among mostly red pigment, noteworthy was a crayon of green copper ore pigment from ARN3 T2

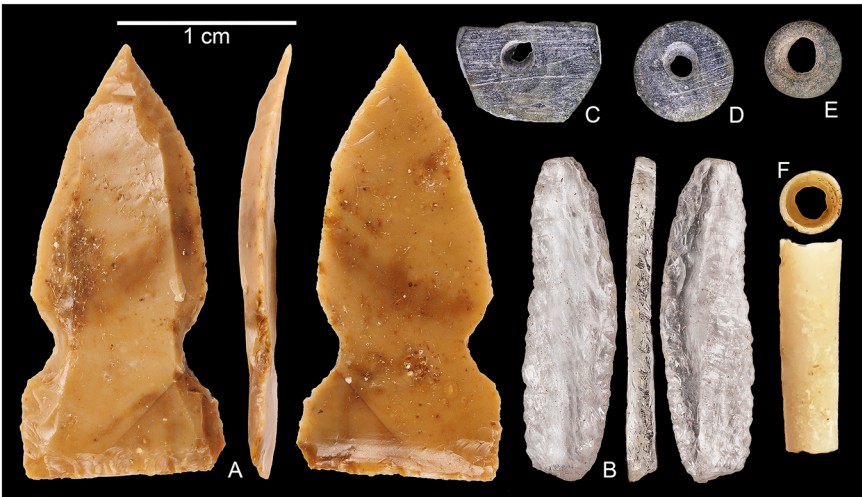

**Fig. 5 | Stone tools and beads recovered from excavations. A** El Khiam point from ARN3; **B** Helwan bladelet from ARN3; **C–E** Ground stone disc beads from the ARN3 T2; **F** Dentalium bead from ARN3 T2. Photos by Antonio Reiss.

layer 7. Green pigment and ground stone beads are characteristic of the earlier PPN[37,38]. Dentalium shells are known from the PPNA[39] and had to be collected from the Mediterranean Sea or the Red Sea, making the latter the closest potential source at over 320 km away.

Three tools with clear battering marks were recovered from directly below engraved panels. One is from a stratified and dated archaeological context underneath a life-sized camel engraving at ARN3 T1 (Figs. 4 and 6), immediately below OSL sample ARN-T1-2, dated to 12.2 ± 1.4 ka, and above sample ARN-T1-3 dated 12.8 ± 1.1 ka (Table 1). Two additional tools were recovered on the surface beneath panels ARN30 and JMI8 (Supplementary Fig. 37). The ARN tools are made on clasts of ferruginous sandstone, a tough material not immediately available at the rock art localities, while the JMI8 tool is made of silcrete, a material available at SAU. The tools are all wedge shaped, and the two made on ferruginous sandstone were flaked to sharpen the pecking edge. The tools comfortably fit into a medium-sized adult hand, with the battered edges protruding (Fig. 6B, C). Battering is evident on multiple edges of all three tools, indicating they were used extensively. Due to their angular form, these tools are not viable as lithic hammerstones in the arced free-hand percussion motion; however, the tapered ends would make good surfaces for the direct end-on percussion required to peck petroglyphs (Supplementary Note 3.3). The tool recovered from ARN3 T1 was used to peck and could thus have been involved in the production of either of the two camel engravings above (Fig. 4). In Arabia, this type of tool was first noted at the Camel Site, where traceological analysis on silcrete pieces showed damage consistent with sandstone engraving[40]. Comparable tools and patterns of wear have also been observed at petroglyph sites in Europe[41] and South America[42,43].

## Discussion
### Archaeology and environment
Archaeological and palaeoenvironmental research provides new insights into the human occupation history of northern Arabia (Fig. 7). An absence of sites dating between the LGM and ~12 ka has been observed across the Arabian Peninsula and was likely exacerbated by rising sea levels, which would have submerged any sites associated with coastal refugia[44,45]. Our evidence for the onset of playa sedimentation indicates that as the hyper-arid climate of the LGM gave way to gradually increasing humidity, ephemeral water bodies were established after ~16 ka at JMI and from ~13 ka at ARN. Human groups in northern Arabia began to exploit these newly forming seasonal water bodies in dryland ecosystems during the Pleistocene-Holocene transition.

Sediment analyses indicate seasonal waterbodies in a landscape that was likely slightly wetter than it is today but too arid to allow the establishment of more permanent lakes (Supplementary Note 5). The fauna represented in the rock art supports this evidence with the identifiable engravings dominated by arid-adapted species, i.e., camels (72%), ibex (12%), gazelles (5%), and wild equids (10%) (Supplementary Data 1). In this context a single depiction of an aurochs at ARN3B (Fig. 2C) stands out and suggests that either these obligate drinkers were able to penetrate the Arabian interior during exceptionally wet seasons or that aurochs were seen elsewhere, perhaps when human populations retreated to wetter areas during the dry season. Currently, the rainfall in the region falls predominantly in the winter months. Observations of modern feral camels indicate their mating season correlates with the wet season[46]. The depiction of male camels in rut[29,30] and with visible winter fur in these engravings[27] thus indicates rainfall may have also coincided with the winter months around 12 ka.

### Monumental rock art
Two independent lines of evidence provide a chronology for the rock art at Jebel Arnaan. Firstly, life-sized engravings of animals are the principal rock art tradition at the ARN localities, with no later or superimposed images present until the introduction of firearms (Supplementary Fig. 5). This indicates that the engravings must have been produced at the time of the archaeological occupation, with the main phase in both ARN3 trenches overlapping in age. Secondly, the lower parts of the engraving behind ARN3 T1 are covered by layer 3, indicating that the images must have been completed by the time the deposition of layer 4 ended. The ferruginous pecking stone in layer 5 would be effective in engraving sandstone. In the absence of other rock art within 200 m of the site, our parsimonious assumption is that the pecking stone was used to produce one of the engraved figures at ARN3, and that corresponding chronometric dates can thus be used to date the rock art. Nevertheless, the possibility that the engraving tool was transported from a different panel at Jebel Arnaan cannot be excluded entirely. Luminescence and radiocarbon ages obtained at ARN3 indicate that the pecking stone was deposited at the Pleistocene-Holocene transition, and the main phase of occupation is contemporary with the PPNA. Our resulting correlation is that at least one of these large animal engravings was produced during the PPNA. This is supported by analyses of rock varnish which have shown that complete re-varnishing of engraved lines (Figs. 2 and 4 inset; Supplementary Figs. 2 and 3) requires ca 8 ka[31,32,47], and match previous observations at SAU and at the Camel Site that this type of engraving

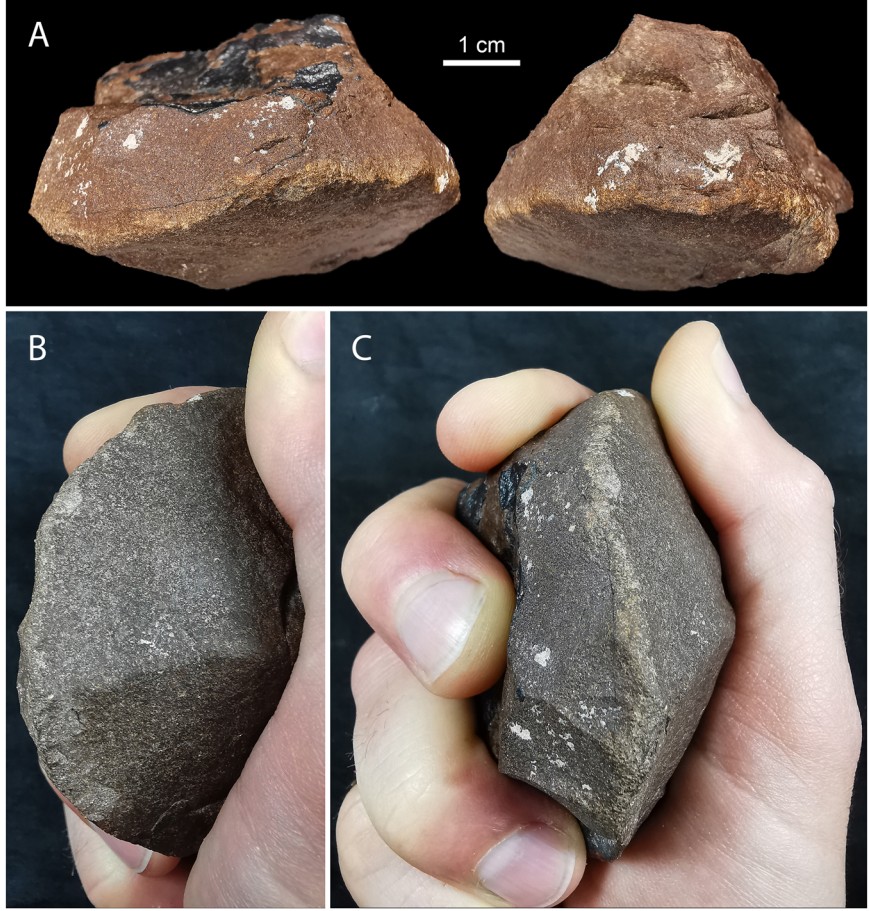

**Fig. 6 | Pecking tool from ARN3 T1, layer 5. A** Two sides of the pecking tool. Scale is 1 cm. **B, C** pecking tool held in a hand. Note the battering on the semi-circular protruding edge. For additional photos see Supplementary Figs. 38 and 39.

does not include any depictions of livestock and thus likely pre-dates the emergence of pastoralism in northern Arabia[27,29,48].

Two earlier rock art phases were recorded with repeated super-imposition by life-sized camels, but these do not have direct associations with archaeological deposits. Small, stylised depictions of women with accentuated curves (rock art phase 1), and subsequent depictions of large, stylised human figures (rock art phase 2, Fig. 2D). These human figures consequently pre-date engravings such as those at ARN3, and are thus likely older than 12 ka (Fig. 7). Similarly, an Epipalaeolithic date had previously been tentatively suggested for stylistically similar figures in the Jubbah oasis based on a relative chronology of rock art superimpositions[25].

Re-engraving events such as the one documented at JMI18 (Fig. 3C) indicate that the tradition of large, naturalistic engravings was long-lived. For new lines to be visible, previous representations must have been sufficiently faded through varnish accumulation, weathering or erosion. This pattern was also observed in the rock art of SAU[27] and appears to be typical for this monumental rock art tradition. Engravings of phase 4, which are often found superimposed over earlier naturalistic depictions of phase 3 (Fig. 2B and 3C), are more cartoonish and follow a more standardised ideal of beauty, reflecting a stylistic evolution over time (Supplementary Note 1 and Supplementary Fig. 4).

During the terminal Pleistocene and earliest Holocene, human populations appear to have moved along established routes determined by a network of water sources, which were marked with monumental rock art. Such 'freshwater corridors' also connected the Nefud and Levant during the much wetter Pleistocene humid events when large permanent lakes formed in the region[10]. At JMI all panels

flank the edge of a playa (Fig. 8C and Supplementary Fig. 6). At ARN, all life-sized rock art panels follow a gulley up the slope in which water pools even today after infrequent rains (Fig. 8B). This route provided convenient access to ephemeral water pools and it would also have acted as a shortcut across the mountain to reach two playas on the western side (one of which is Site 1, ARN), where water would likely have been available for longer periods after infrequent flood events (Fig. 8A). All panels with life-sized animals overlook this route and would have been highly visible when freshly engraved (Supplementary Fig. 1).

The engravings express a symbolism that relates to desert animals and to seasonality, namely the depiction of male camels in rut, which may have referenced the mating season of these animals after the winter rains[46], and potentially also their extraordinary resilience in these arid landscapes. In the drylands of the Pleistocene-Holocene transition, hunter-gatherer groups were likely highly mobile. Depictions of wild animals and the marking of natural landscapes may have been a mechanism to mark routes and perhaps also served as territorial markers documenting access rights via impressive images that would have "guarded" locations in a group's absence. The engravings, which may have been created over a time span of millennia, would have reminded people of ancient symbolisms and beliefs of their group, which likely structured their highly seasonal lives and thus enhanced their ability to thrive in these marginal landscapes.

## Levantine connections

Mounting archaeological evidence suggests repeated contact between human populations in northern Arabia and the Levant, throughout the terminal Pleistocene and the Holocene. Arrowheads associated with

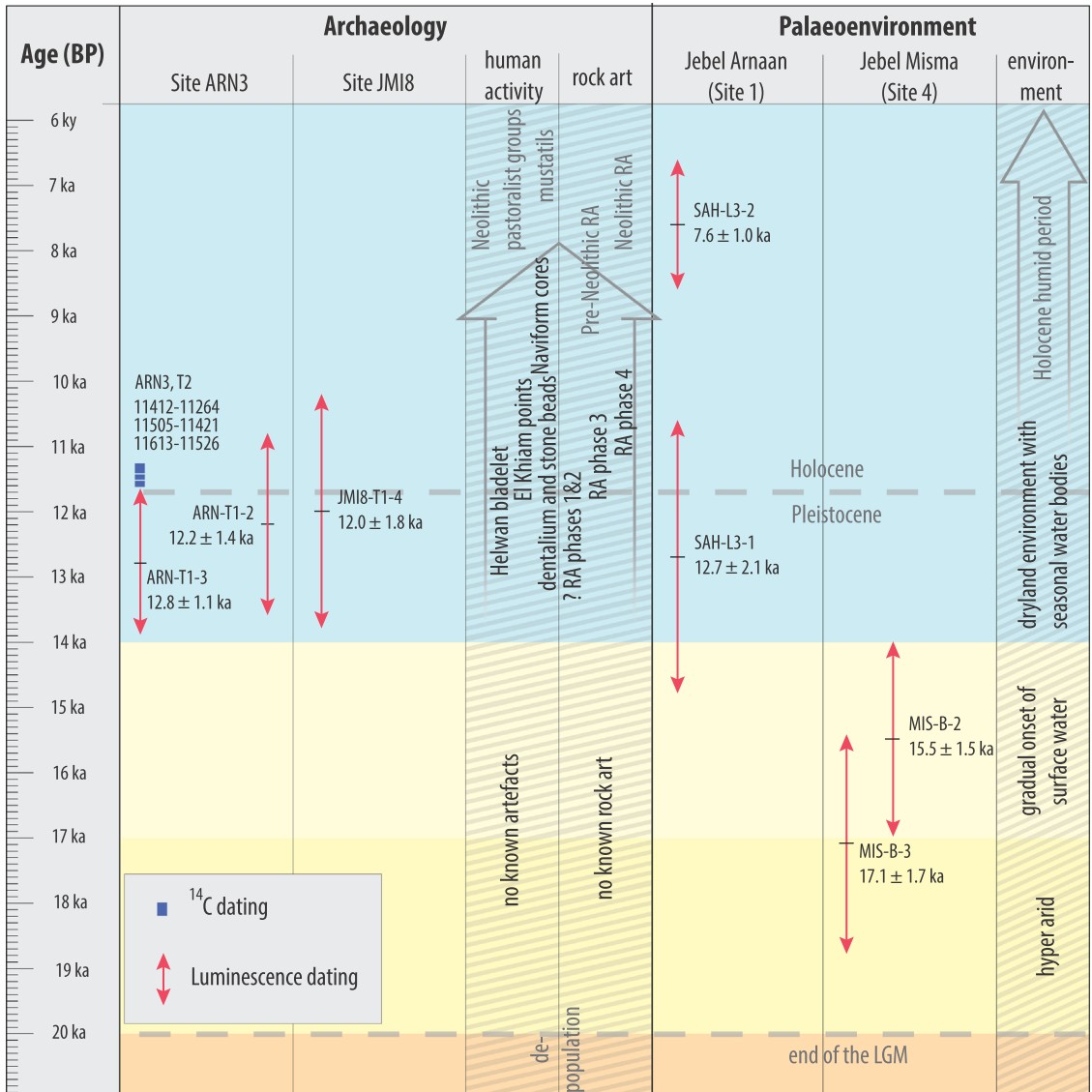

**Fig. 7 | Compilation of radiocarbon and luminescence ages of the playa sediments and archaeology.** 2σ 14 C age ranges are presented in calibrated years BP, while luminescence age ranges are shown at 1σ in ka prior to the measurement datum of 2023. Evidence for human activity and environments listed in grey refer to published research[11,19,22,57], evidence listed in black is presented in this paper. Artefacts listed under human activity were recovered during excavations at ARN and JMI and are dated based on OSL and 14 C ages recovered at ARN and JMI, as well as typological comparisons with the Levant. The timing of rock art phases reflect the interpretation of our evidence presented in the text: Rock Art phases 3 and 4 are based on OSL dates from a layer containing a pecking tool at ARN T1, a radiocarbon age from ARN3 T2, an OSL date from JMI8 T1, and the overall characteristics of artefact assemblages at ARN and JMI. The timing of Rock Art phases 1 and 2 is estimated based on superimpositions. The environmental phases are based on a change in the balance between aeolian erosion and fluvial sedimentation that is visible in the sedimentary record, with increasing fluvial sedimentation indicating a reduction in aridity.

the Late Neolithic in the Levant, such as Ha-Parsa, Nizzanim, and Herzliya types, as well as decorative artefacts such as shell beads and stone bracelets, have been identified at Jebel Oraf[19] and at multiple sites in the AlUla region[49]. PPNB technologies such as naviform cores are well documented from the Jawf region north of the Nefud desert[50], with the discovery of Helwan points at Jebel Oraf[51] extending these links into the southern Nefud desert, where PPNA technology in the form of El Khiam points is also known from Jebel Qattar[18] in the Jubbah oasis. The recent discovery of a Helwan bladelet at SAU, alongside a radiocarbon age of 13.4 ka indicates that these connections may reach back to the terminal Pleistocene[27].

The artefact assemblages presented here include Levantine stone tool types and decorative forms of material culture (green ground stone disc and dentalium beads), suggesting regular and repeated cultural contact across long distances. These contacts are evident in the initial occupation of the region at the Pleistocene-Holocene transition, and continued into the early Holocene, with PPNA and PPNB cultural entities represented in the material culture. Luminescence and radiocarbon ages obtained from stratigraphic contexts indicate that the use of these artefacts was contemporaneous with the Levant (Fig. 7). To our knowledge, this is the first time PPN artefacts have been found in a stratified and dated context in Arabia, and the first time they have been associated with this style of monumental naturalistic art.

The distribution of the monumental rock art along routes connecting seasonal water bodies suggests high mobility, and regular movements across considerable distances may have been a key adaptation to the cyclical availability of water. In one case this may even have been captured in three rock art panels north and south of the Nefud desert, which appear to have been re-worked multiple times by the same succession of artists[48]. Connectivity is also a key feature of

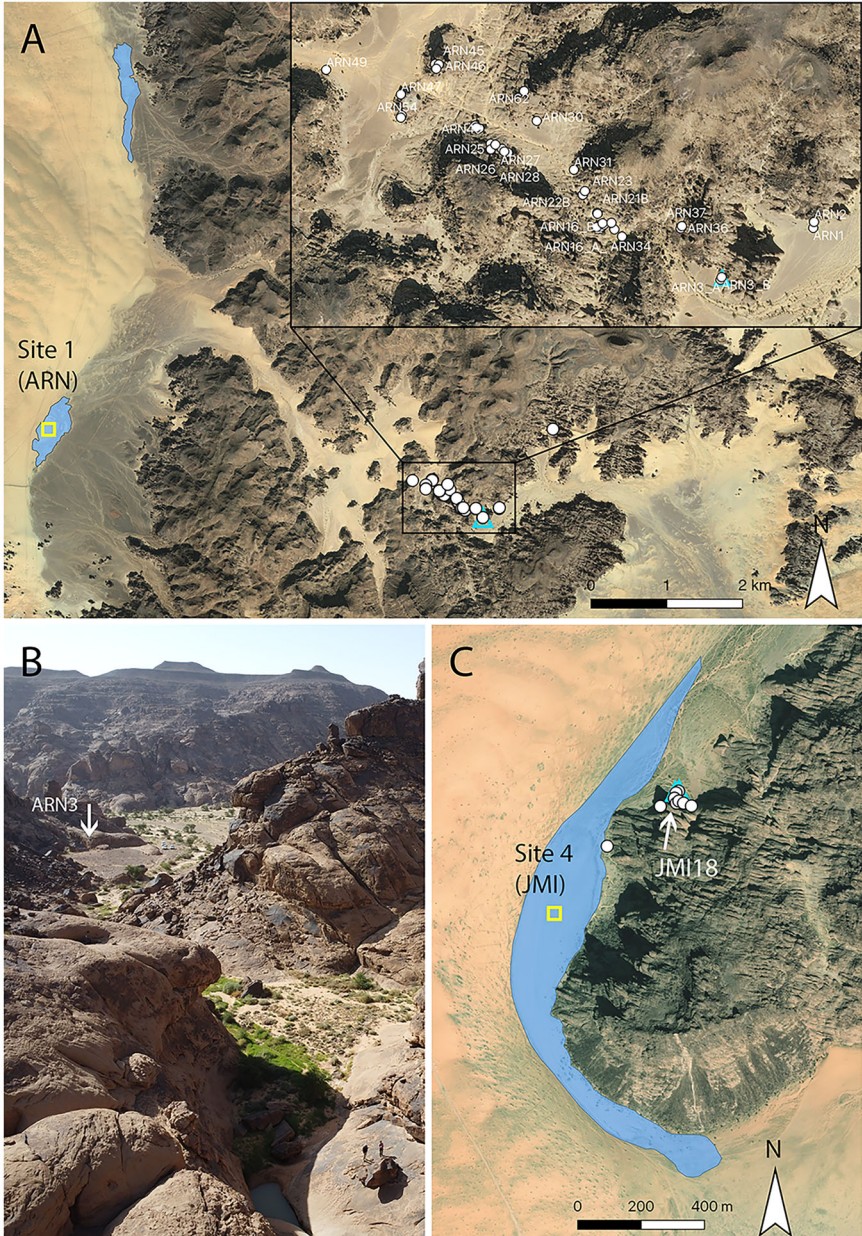

**Fig. 8 | Palaeolakes and modern landscapes at Jebel Arnaan and Jebel Misma.**
**A** Palaeolakes on the western slope of Jebel Arnaan. Rock art panels follow a gully up the mountain that connect to a valley which leads to two palaeolakes. Excavation of lake sediments marked in yellow, archaeological excavation marked with a blue triangle. Inset: close up of the rock art distribution at ARN, showing all panels with large animal engravings along a gulley in which water pools today. **B** View down the slope towards the ARN3 excavations (arrow). Note the two vehicles between the arrow and the wadi channel for scale. In the foreground water can be seen pooling in a rock pool. Note the two people standing to the right for scale. Rock art follows the water course up the slope. **C** Palaeolake and life-sized rock art at Jebel Misma. Panels were located on the edge of the palaeolake and overlooking the northern end of the lake where the terrain forms an embayment between the lake and the jebel slope (e.g. JMI18). A water hole directly east of the mapped panels held water during excavations in May 2023 (Supplementary Fig. 8). Bing Virtual Earth imagery as basemap in QGIS. Imagery © 2025 Microsoft Corporation.

Epipalaeolithic occupations in the Levant and Jordan[52], with such connectivity increasing in scale in the PPN[53,54]. Population movements in both northern Arabia and regions further north may therefore have facilitated the transfer of knowledge while also accommodating distinct cultural expressions.

The interdisciplinary geoarchaeological fieldwork at Sahout indicates that the populations who engraved the monumental rock art panels of Jebel Arnaan and Jebel Misma were the first occupants in the interior of northern Arabia since the LGM (Fig. 7). These pioneers were able to thrive in the arid conditions of the terminal Pleistocene and earliest Holocene due to seasonal water bodies. The presence of key

lithic artefacts such as El Khiam and Helwan points, as well as decorative artefacts such as green pigment and dentalium beads, suggests these human groups maintained contact with their Levantine neighbours during the PPN, travelling across vast distances. However, the engravers of Jebel Arnaan and Jebel Misma had their own, distinct cultural and symbolic identity. Their adaptation to an environment where water was only available temporarily, involved complex mobility along routes connecting different water sources. Unlike their Levantine neighbours, they produced monumental rock art that centres around a desert animal symbolism: the camel. These monumental images were used to mark water sources and the routes between them,

perhaps providing impressive visual reminders of access rights, while also commemorating these extraordinary desert-adapted groups over millennia.

## Methods

### Permits and permissions
Permission for excavations in Saudi Arabia, sampling, export of samples and artefacts to the United Kingdom, and analysis was granted by the Heritage Commission, Ministry of Culture, Saudi Arabia.

### Rock art
Rock art panels were recorded using pro-forma record sheets, with content, size and GPS location noted, and documented using high-resolution digital photographs as raw files and JPGs. To enhance faded details on the rock art, some photos were enhanced using Adobe Lightroom. Key panels were photographed systematically to allow generation of high-resolution 3D models and orthophotos using Metashape photogrammetry software. One exceptionally large panel was located on a cliff 34–39 m above the ground (Fig. 3C) and had to be documented using a DJI Mavic Mini 2 UAV due to its height and inaccessibility, with the resulting photographs combined into an orthophoto to facilitate analysis.

Rock art surveys were carried out systematically, with survey teams following the base of jebels and exploring all boulders and rock surfaces along the base and within heights that could safely be reached. At Jebel Arnaan, rock art survey followed a gully that leads up the mountain slope and connects to a wadi above (Fig. 8B).

A total of 106 rock art panels were documented across all three regions (Jebel Arnaan, Jebel Mleiha and Jebel Misma). Distinct surfaces of boulders were defined as separate panels, with multiple surfaces on the same boulder labelled with the same number, distinguished by a letter suffix (e.g. ARN3_A, ARN3_B).

### Excavations
All trenches were hand-dug by trowel using the single context method[55]. Thicker deposits were sometimes divided into multiple contexts for extra stratigraphic control. Contexts were recorded using project-specific pro forma record sheets. Depths were recorded with a line level, with five points taken for each context. Excavated sediment was sieved through a 3 mm mesh with all artefacts and faunal remains retained. The position of key finds was manually measured using a plumb bob from an origin in one corner of each trench.

### Artefacts
All lithics were classified by material, counted, and weighed. Any retouched pieces and cores were assigned to types. Key types were photographed and measured. Grinding stones were assigned to types, their ground surface form characterised, and any complete dimensions measured. Pigment was weighed and pieces with striations visible at low magnifications designated as crayons. Bead diameter, aperture width, and thickness were measured.

### Luminescence dating
Luminescence dates presented in this study were determined using optically stimulated luminescence techniques applied to quartz. These measure the time elapsed since sediments were last exposed to sunlight. Opaque metal tubes were hammered into cleaned sections, transported to the UK and analysed as described in Supplementary Note 6.1. Environmental dose rates were calculated using location and overburden density (cosmic rays), field gamma spectrometry (gamma), and thick-source beta counting (beta). Luminescence analyses were carried out by Simon Armitage at the School of Life Sciences and the Environment, Royal Holloway University of London, United Kingdom.

### Radiocarbon dating
Samples suitable for radiocarbon dating were collected from stratified archaeological deposits during excavation and included charcoal-rich sediment samples from hearth deposits, ostrich egg shell, micro charcoal, and one piece of shell. The latter was identified as a pearl oyster (Supplementary Note 4). Samples for radiocarbon dating were sent to the Centre for Applied Isotope Studies (CAIS) at the University of Georgia and analysed as described in Supplementary Note 6.2.

### Palaeoenvironment
Trenches were dug in the middle of the playa settings, up to 2 m deep. The sections were described sedimentologically and sampled for geochemical lab analysis and luminescence dating. The XRD semi-quantitative analysis of the major mineral components was carried out using a D8 Bruker machine, at the laboratory in KAUST, with a chosen angle ($2\theta$) between the incoming and outcoming X-ray beam between 5 and 80°. The mineral phase interpretation was done using DIFFRAC.EVA software and the results are presented as a proportional percentage of the mineral phase present in the analysed samples (Supplementary Note 5). The samples were powdered using a handheld mortar.

### Reporting summary
Further information on research design is available in the Nature Portfolio Reporting Summary linked to this article.

## Data availability
All data used in this study are available in the Supplementary Information files. All data used to support our results were generated in our research. Sediment samples from playa deposits are stored at King Abdullah University for Science and Technology, Physical Science and Engineering Division. Artefacts recovered during excavations will be returned to and stored by the Heritage Commission, Ministry of Culture, Saudi Arabia, in 2025. Access can be negotiated with the Heritage Commission (faljibrin@moc.gov.sa).

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

## Acknowledgements

We thank HH Prince Badr Bin Abdullah Bin Farhan Al-Saud, Saudi Minister of Culture, for permission to conduct research at Sahout. We also thank Dr. Jasir Alherbish, CEO of the Saudi Heritage Commission, and Dr. Abdullah al-Zahrani, General Director for Archaeological Excavations. Financial and logistic support in the field was provided by the Saudi Heritage Commission. Funding for fieldwork and research was provided by a British Academy/Leverhulme Small Research Grant (SRG2223 \231473 to M.G. and C.S.). We acknowledge funding from King Abdullah University of Science and Technology (KAUST) from baseline support of F.v.B. S.J.A. contribution to this work was partly supported by the Research Council of Norway, through its Centres of Excellence funding scheme, SFF Centre for Early Sapiens Behaviour (SapienCE), project number 262618. A.M.A. acknowledges the support of the Researchers Supporting Project number (RSP-2025/126), King Saud University, Riyadh, Saudi Arabia. We also thank Saleh Idris, a labourer who assisted with project excavations, and who discovered panel JMI18.

## Author contributions

M.G.: conceptualisation, data curation, funding acquisition, investigation, methodology, visualisation, writing—original draft preparation, writing—review and editing. C.S.: conceptualisation, data curation, funding acquisition, investigation, methodology, visualisation, writing—original draft preparation, writing—review and editing. F.A.-J.: funding acquisition, investigation, resources. G.L.: Data curation, formal analysis, investigation, methodology, visualisation, writing—original draft preparation. A.K.: data curation, formal analysis, investigation, methodology. S.J.A.: data curation, formal analysis, investigation, methodology, writing—review and editing. F.S.: data curation, investigation, methodology. M.S.: data curation, formal analysis, investigation, methodology, writing—review and editing. F.A.-T.: investigation. P.S.B.: data curation, formal analysis, investigation, methodology. F.v.B.: data curation, funding acquisition, formal analysis, investigation, methodology, writing—review and editing. N.D.: data curation, investigation, methodology, writing —original draft preparation, writing—review and editing. M.A.-S.: investigation. A.A.-S.: investigation. J.A.-W.: investigation. A.M.A.: investigation, writing—review and editing. MP: investigation, writing—original draft preparation, writing—review and editing.

## Funding

## Competing interests

The authors declare no competing interests.
