## [Transparent Peer Review file · Nature Communications]

Monumental rock art illustrates that humans thrived in the Arabian Desert during the Pleistocene-Holocene transition

Corresponding Author: Dr Maria Guagnin

Version 0:

Reviewer comments:

Reviewer #1

(Remarks to the Author)

With their paper „Monumental rock art illustrates that humans thrived in the Arabian Desert during the Pleistocene-Holocene transition“, Guagnin and colleagues provide data on rock art, paleoenvironment and archaeology of the terminal Pleistocene to earliest Holocene period from northern Saudi Arabia. In a spatial extension of their previous study Guagnin et al. (2023, *Archaeological Research in Asia* 36) they demonstrate here the presence of life-size camel engravings and archaeological records broadly similar to their previous findings about 15 km to the north. However, the authors state here that: “For the first time, secure stratigraphic contexts from excavations directly beneath the rock art panels allow these engravings to be dated.” (page 4 main ms, 2nd paragraph).

In general, the results presented are interesting. However, I argue below that only the archaeological results provide somewhat solid data for publication. The arguments and data that the authors present with regard to the dating of rock art and their reconstruction of palaeoenvironmental conditions, in contrast, are in my opinion not supported by the presented data. I detail my concerns below. Since they are substantial, I will not go into the details of the minor issues.

In its current state, I cannot recommend publishing this manuscript in Nature Communications.

One of the main topics in this paper is the age of the life-sized camel engravings. It is always a challenge to date engravings and having sediment deposits partially covering these is one of the more promising options to get a minimum age. The authors present one example of such a context (trench 1 in their ARN3 area). Here they link OSL ages from layer 5, about 40-50 cm below the camel engraving with the engraving. This excavated layer has multiple lithic artifacts and in particular one engraving tool could be found in that layer. I understand that the authors would like to see a link between engraving and engraving tool given the spatial proximity, but from a strictly scientific and stratigraphic perspective this conclusion is not supported by the data (see figure 3). There is no stratigraphic connection between the dated OSL samples and the engraving. The only possible conclusion is that these engravings are younger than the age estimations of layer 5. Whether this is a couple of years, a few decades or millennia, remains speculative.

Considering rock art, I noted that the camel shown in grey in figure 3 has its hump very far back. To me this does not look like a normal dromedary that one would expect as part of the local wild fauna. I was wondering if this could then be a Bactrian camel with two humps or an interbreed between dromedary and Bactrian (in Arabia called “bukht”)? Bactrian camels did not occur in Arabia naturally and were only imported during the Iron Age. The authors should provide additional information on the camel species and should consult an expert for camels. I further noted that the lower parts of the legs in figure 3 are shown with dashed lines. What does that indicate? Were the engravings incomplete here? This also needs to be clarified since this is important when assessing the spatial relationship between engraving, the sand deposits and the OSL sample locations.

The archaeology and its age estimations on the other hand are comprehensible and well demonstrated. Although for the presentation of new archaeological material I miss a detailed description of how the excavations have been conducted. Did the authors sieve for example? How did they remove the sediment, all area at once or did they subdivide the excavated area as it would be a modern standard procedure? How did they document the position of the finds within the excavated layer? In Suppl. Fig. 11 there is no obvious instrument that would allow measuring the position of the findings, so how have the authors done the documentation? Moreover, what does for example “layer 5 (context 006)” as mentioned in the suppl mean? What is the context in this case? The counts on lithics provided, what do they include? How many lithic artifacts larger than 2

cm did the authors document and how is their horizontal distribution? Please avoid expressions such as “many” or the “majority” when presenting numerical data. Provide numbers instead to allow the reader to follow your presentation. So, while I consider the archaeology relevant, the presentation of the archaeological excavations and findings require significant revision following my questions above.

Since I am not a specialist, I cannot comment on the presented chronometric dating results and related methods.

Beside my concerns regarding the age of the engravings, the other major issue is the scarcity of convincing evidence provided by the authors for the claim of “more humid conditions with increased water accumulation” (page 6 of the main manuscript) demonstrated by the “prolonged sequence of playa deposits” (ibid). Playa deposits is a flexible term that I think needs to be defined. Given table 1 and the grain sizes listed there, all but one context is composed of fine sand, which might be washed or blown in. In my understanding this is not the classic type of sediment that forms exclusively in standing water bodies. I think the reader would much appreciate it if the authors would define what they consider to be playa deposits exactly and how they differentiate between different depositional conditions.

Reading through their description of the palaeoenvironment results left me somewhat confused since there are contradictory statements. On the one hand they say: “[...] subordinate carbonate concentration, and the absence of root traces and organic-rich layers suggests that conditions remained too dry for the establishment of more permanent water bodies, indicating that a dryland environment persisted [...]” (page 6 of the main ms). While only a few sentences later they say: “These sites thus represent the earliest evidence from northern Arabia of increased humidity following the hyper arid LGM. However, the lakes were ephemeral, indicating an arid or semi-arid climate prevailed in the region.” (page 7, main ms). So what is the author’s final conclusion? To me the presented data clearly speaks in favor of a continuity of arid conditions with infrequent rain events that led to the formation of some shallow puddles, which have existed for a few days. A phenomenon that can also be observed today in Arabia and other deserts after rain. If there is no further supporting evidence for a longer duration of water bodies from these sections, such as the presence of pointer species for lacustrine conditions, these deposits can in my view not be cited as evidence for increased humidity.

To wrap up, I think the presented conclusions for “securely dated rock art” and “the earliest evidence for increased humidity from northern Arabia” are not supported by the presented data. While saying this, the presented archaeological data is, although not comprehensively presented, at least convincing with regard to the chronology and cultural context of the population settled here at some time between 13.8 ka and 10.9 ka ago. That the authors can link this presence to Levantine Khiamian tradition is very interesting and would be worth further exploration. The excavations are relatively small since I suppose they have been started initially to collect chronometric data for the rock art. It might be worth going back, open larger trenches and see what the groups carrying Khiamian traditions else have left in the deep desert. The 500+ artifacts (if not 490 were smaller than 5 mm, please clarify the frequency of artifacts larger than 2 cm) could hint at an archaeological record of decent size. Finally I would like to point out that the dating of the Khiamian material from this site is relatively early compared to the standard time range proposed for the Levantine sites. Again an interesting detail that could be further explored.

Reviewer #2

(Remarks to the Author)

In this article, the collaborating authors draw upon multiple strands of archaeological and paleoecological evidence to reconstruct a late Pleistocene-Early Holocene environment and populate it with fauna and human groups. For the latter, the archaeological evidence offers chronological and some cultural details such as technical affinities with northern traditions and long-distance exchanges of exotic materials. Mobility of these human groups is implicit, by virtue of the seasonal availability of local resources and potentially by depictions of seasonal characteristics of fauna. A major focus of this paper is the local rock art, studied in three newly reported locations and linked to well-reported rock art studies across a broader region of northern Arabia. While rock art is rich in imagery and clues to the symbolism and communications of its creators, it is rarely possible to date it or to link rock art unequivocally to other details of the archaeological record. The significance of this paper lies in the assertion that the earliest phase of this reported rock art can be securely assigned to the Late Pleistocene-Early Holocene transition and that the reported artifacts show an expansion of people into formerly hyper-arid zones after the Pleistocene LGM abandonment. This argument supposes that rock art of Phase 1 serves as a marker of human activity earlier than previously documented Late Pleistocene-Early Holocene Arabia.

There are two key strands to this argument for which the authors supply details and which this reviewer is best qualified to assess. (Paleoecological details of sediment characteristics and dating methods underpinning the regional environmental reconstructions, including paleolacs, are not my forte). One strand is the age of the rock art and its indirect dating through purported association with artifacts. The second strand is the absence of evidence for Late Pleistocene and Epipaleolithic occupation and use of local resources to purport a re-population/expansion, driven by exogenous climatic factors) into a region previously lacking human occupation. This second strand is the major justification for publication in Nature Communications, even as the evidence and topic of rock art is intrinsically interesting, if localized. Similar and intrinsically localized evidence can be of the highest scientific interest, as studies of 40,000-year-old cave paintings in Southwest Sulawesi have demonstrated.

To examine these arguments, therefore, some close consideration of the linkages and chronologies of rock art and excavations is warranted. While the evidence is well presented, very well illustrated, clearly organized, and amply supplemented with extensive online supplemental materials, there remain some ambiguities. It is fortuitous to find sediments sealing or obscuring rock art as was the case at site ARN3 T1. This is the only fully reliable stratigraphic dating (indirect) for rock art.

A critical item therefore is the white camel (phase 1) on the panel ARN 3 with its legs covered in sediment. The authors assign this to the Pre-Pottery Neolithic (layer 5 in ARN T1). With a rockfall (layer 4) overlying layer 5, the camel engraving (and all phase 1 rock art classed as contemporaneous with it by the authors) could be later than layer 5, engraved independently and sealed at a later date by rockfall. While the evidence is strongly suggestive of a PPNA (not Natufian) association with layer 5 and therefore of the rock art “engraving” of a camel, questions remain. The tool argued to be used for engraving is also described as a “pecking stone” (Supplement 3.3 p. 16-170, and this pecking motion is also described in Supplement 3.4 p. 42 (Figure 37 on p. 44). I would like to see clarification of

A) The reasons for dating to Natufian (a few notched flakes and mortars) instead of PPNA (later, consistent with and unremarkable for other finds in Arabia, e.g., Jebel Oraf). Note another “pecking” tool recovered on the surface at ARN30 (rock art production method not specified in Supplementary Table 1). Radiocarbon dates are consistent with PPNA, not Natufian—unless one reads the most ancient extent of the radiocarbon determination rather than its median.

B) The difference between engraving and pecking and the reasoning for a pecking tool to be an engraving instrument. This is critical for the association of the tool in ARN3 T1 and the camel engraving above it. I suggest more text on how the rock art was produced (pecking, engraving, other?) and explanation whether production criteria was included in the analysis of phases/chronology—or do different production techniques cross-cut the chronological assignments of phases? I also suggest (in light of the access challenges) that authors address whether one or many people were needed and consider the person hours of labor involved. These considerations may lead to further insights on group size, frequency of passage, and community dynamics.

C) The reason the camel engraving cannot be later than layer 5, that is, PPNA occupants at layer 5. As I read their evidence, it is plausible that there is a abandonment layer, then layer 4, which covers the camel engraving of rock art phase 1). It is possible that the camel engraving postdates layer 5 (by an unspecified period of time) and pre-dates layer 4 collapse (documented by authors in section Figure). It would seem that a post PPNA composition of phase 1 would be very much consistent with evidence elsewhere of PPNB cultural influences among populations in the north Arabian humid period.

Note that if the evidence is stronger (as I read it to be) for PPNA not Natufian incursions into the arid zone, the chronological assertions are less remarkable than the authors indicate, or they could be re-stated for local versus peninsular significance. (See Crassard et al. 2013, cf., Hilbert et al. 2014 both cited by authors). Even so, the analysis classifying different episodes of rock art based on stylistic grounds and overlap/superposition of images is new and potentially foundational in the region.

The second strand is the absence of evidence for Late Pleistocene and Epipaleolithic occupation (at this locale) and use of local resources to purport a re-population/expansion, driven by exogenous climatic factors. The authors need to broaden the significance of this paper beyond the repopulation of this specific locale in the Arabian Peninsula. This might be done by referencing other and adjacent geographies: the greening of the Sahara or the LGM refugium in the Arabian/Persian Gulf theory come to mind.

A few minor points:

Caption in Figure 1 (main text) documents 131 rock art images, versus (in text) account of 176 images (line 150).

Line 234—dating the engraving of the camel by the height of its legs. Even if the authors’ reasoning is sound (? See above), this line is confusing and needs to be clearer for first reading.

Line 297 and ff. lithics—Supplement p. 36 & Fig 20—that there is a transition in materials between PPNA and PPNB layers is interesting and deserves greater comment. A gap in time? Different populations? Different economic pathways for the circulation of goods?

Line 379 argues that engravings reflect a rainy season. I tend more to agree with the multiple and sometime symbolic/ideological and/or social communications the authors explore in discussion. It is altogether too easy to “read” as economic and functional-historical images one doesn’t access through cultural practice and initiation. I’d prefer to keep the interpretation open and refrain from linking specifically to moisture/rainy season.

Supplementary Figure 10 has a dark shadow that obscures critical section information. The authors would be forgiven for photoshop manipulation of this figure to illuminate any section details that can be discerned.

Likewise it is odd that the authors elected not to excavate fully adjacent/perpendicular to the limestone face bearing the rock art (as they did in ARN3 T2). If this was done subsequent to the image, a description of the lowest extent of phase 1 camel engraving (outlined in black) would be warranted.

Missing references include other efforts at dating rock art in Arabia, either through indirect association with sediments (Rachad and Inizan’s work at Sa’ada, Yemen) or through stylistic classifications and their superposition in different panels (also used in this manuscript) (e.g., Philby-Ryckmans’ expedition publications). Direct dating has recently been possible with (organic) painting (Rowe et al. The Holocene 2022).

(Remarks to the Author)

A very interesting piece of work. It is good to see the archaeology tied to the rock art, together with the various dating methods. A few remarks:

Line 93 What stone are the carvings on (I assume sandstone, but please state at the outset)

Line 160 (Fig. 1) A: Map could be larger and B possibly two gazelle and one goat

Line 205 Freshly engraved INSERT COMMA the images...

Line 366 omit comma

Line 374 could the aurochs not be a record of what has been seen elsewhere?

In the paragraph starting on l. 431 maybe add something about representation of the animals also being a record of the environment, potentially sympathetic magic, territorial markers, indications of herd movements (an extension of the statement about routes l. 437),

Line 489 the monumental rock art in Arabia sets them apart from the Levantines—maybe a phrase to explain what sort of rock art was found in most of the Levant... Also, maybe worth noting that such monumental rock art is found in North Africa, though not in Egypt. And, it might be worthwhile speculating (if possible) why monumental images were so common here.

What animal bones were recovered from the area? Can these be linked to the rock art?

Reviewer #4

(Remarks to the Author)

The manuscript entitled: "Monumental rock art illustrates that humans thrived in the Arabian Desert during the Pleistocene-Holocene transition" presents highly original data from a key archaeological site in Northern Arabia. With regard to the interdisciplinary scope and the relevance of the topic presented in the Ms I have no doubt that this manuscript is a potential very good fit for Nature Communications as it is relevant for the broader scientific community. Also, with regard to the highly original multi-method dataset this Ms deserves publication in a high-ranking outlet. However, I was a bit less impressed with the structure and conciseness of the main manuscript. The manuscript is occasionally redundant and the argumentation is – at least partly – not convincingly structured. Even more importantly it is oftentimes difficult to trace back the scientific evidence that should support the argument/interpretation. Also, some of the figures need to be improved especially with regard to their self-explanatory power. From my perspective it is crucially important that the actual scientific evidence is directly associable with the corresponding figure (especially Fig. 6) or interpretation. In summary, I see the high potential of the paper and I am convinced that it can have a significant scientific impact, however, a major revision is required to make it fit for publication. Below I provide some feedback that will hopefully help to make the Ms more concise. With regard to the supplement, I focused on the geomorphological (note 5) and the luminescence data (note 6) and provide some additional feedback especially with regard to documentation and interpretation.

- please find detailed feedback in the reviewer attachment.

Version 1:

Reviewer comments:

Reviewer #1

(Remarks to the Author)

I thank the authors for considering all my suggestions. The paper is now in a state where I can recommend publishing it.

Reviewer #2

(Remarks to the Author)

This is my second review, and I am grateful for an opportunity to see the changes made to the manuscript as well as the issues raised and responses to other reviewers. I consider this process akin to joining a review panel or scholarly jury after consideration and review of the work by myself. This comment is therefore influenced by my readings of other reviewer-peers and of course by my own re-consideration of manuscript and revisions. I caution that I have provided a more cursory reading than on first review. It is interesting to see that reviewers have raised some of the same concerns and points while bringing different expertises to this review.

I find that the authors have made substantive, highly informed and generally persuasive responses to the many detailed comments we reviewers have raised. There is no doubt that this work is of highest quality, that the methods and analyses are sound, cautious, and address significant questions. Notably, the repopulation and adaptations of human groups to arid environments has significance beyond the data set represented by this small cluster of sites in Northern Arabia.

My lingering concern is the regrettable paucity of evidence that links the first phase (camel images) of petroglyphs to PPNA (the authors have corrected prior text that led me to suppose they argued a Natufian date). The interpretation still hangs on one pecking/engraving stone found 72 cm away near the top of a layer dated by OSL and by a hearth (14C) in a different trench. The excavation is small; as much as I sympathize with the authors' urgency to publish before seeking additional

funding to expand the data set (their letter of reply), I worry that their interpretation still has privileges to an earlier date over potential explanation for a later one. 200 m is not so very far to transport a pecking tool (from a later engraving phase), possibly with the intent to use it, but instead it was lost or abandoned. It is my experience that sedimentation in the desert is neither continuous nor solely depositional. Where material accumulates (layer 5), it can also blow away, leaving heavier stone on the surface (desert reg). This can happen multiple times; thus what appears to be the upper part of layer 5 can reflect an accumulation of artifacts of different ages and sources. Again, the evidence would be more compelling if the hearth was nearer the camel, if the camel legs were clearly complete, if there were more pecking stones or it sat in a dated hearth, or if the trench were extended to join the trench where the dated hearth occurs.

Archaeologists cannot always obtain the richness of data they would prefer—discovery relies on chance and expertise. My point is not to discredit the excellent work the authors have done, and their high expertise is evident.

Reviewer #3

(Remarks to the Author)

The revised work reads much better and address all the reviewers' concerns.

I agree with the authors that the artefacts for creating the images would be beneath (maybe well beneath) the images, though it is always tricky with rock art to always link archaeology to particular works.

It is hoped that more funding is forthcoming so that the authors can return to the site and extend the area of excavation.

Reviewer #4

(Remarks to the Author)

I would like to thank the authors for their detailed rebuttal. As a result of I think that the revised manuscript is much improved and that my main concerns are sufficiently addressed. I am happy to recommend the Ms for publication in Nat.Comm.

REVIEWER COMMENTS

We thank the reviewers for the time they have taken to review our manuscript. We appreciate their detailed comments, which have helped us clarify and strengthen our manuscript. We have addressed all comments (see below, in blue) and have made extensive changes to the text and figure captions, we have included six new references, provided an additional figure, and changed four figures in the manuscript. The Supplementary Information is now 587 words longer and we have added 1 new figure.

Reviewer #1 (Remarks to the Author):

With their paper „Monumental rock art illustrates that humans thrived in the Arabian Desert during the Pleistocene-Holocene transition“, Guagnin and colleagues provide data on rock art, paleoenvironment and archaeology of the terminal Pleistocene to earliest Holocene period from northern Saudi Arabia. In a spatial extension of their previous study Guagnin et al. (2023, *Archaeological Research in Asia* 36) they demonstrate here the presence of life-size camel engravings and archaeological records broadly similar to their previous findings about 15 km to the north. However, the authors state here that: “For the first time, secure stratigraphic contexts from excavations directly beneath the rock art panels allow these engravings to be dated.” (page 4 main ms, 2nd paragraph).

The previous study mentioned by the reviewer consisted of 2 extremely small test excavations (80cmx80cm and 20cmx20cm, to a depth of 12 cm). This scant material only produced a single ¹⁴C predating the Neolithic, and did not provide any data to date rock art. These test excavations were conducted near rock art but not in a location directly associated with rock art (i.e. not directly below an engraved panel – as is the case in the present study at Jebel Arnaan). All we were able to say in this previous paper was that the rock art seems to mirror the occupation sequence and “may have begun earlier than previously thought” and that there was “broader geographical extent to human occupations prior to the Holocene humid period” (Guagnin et al., 2023). We reference this data throughout the manuscript.

We have added a sentence “However, the limited excavations did not permit a correlation between dated deposits and rock art.” To state this difference more clearly.

In general, the results presented are interesting. However, I argue below that only the archaeological results provide somewhat solid data for publication. The arguments and data that the authors present with regard to the dating of rock art and their reconstruction of palaeoenvironmental conditions, in contrast, are in my opinion not supported by the presented data. I detail my concerns below. Since they are substantial, I will not go into the details of the minor issues.

We realise that in our effort to keep the text concise our manuscript lacked research context, particularly on rock art dating and on the formation of rock varnish. We have now integrated this information and thank the reviewer for alerting us to these gaps.

One of the main topics in this paper is the age of the life-sized camel engravings. It is always a challenge to date engravings and having sediment deposits partially covering these is one of the more promising options to get a minimum age. The authors present one example of such a context (trench 1 in their ARN3 area). Here they link OSL ages from layer 5, about 40-50 cm below the camel engraving with the engraving. This excavated layer has multiple lithic artifacts and in particular one engraving tool could be found in that layer. I understand that the authors would like to see a link between engraving and engraving tool given the spatial proximity, but from a strictly scientific and stratigraphic perspective this conclusion is not supported by the data (see figure 3). There is no stratigraphic connection between the dated OSL samples and the engraving. The only possible conclusion is that these engravings are younger than the age estimations of layer 5. Whether this is a couple of years, a few decades or millennia, remains speculative.

Since engravings are produced on surfaces that are clear of sediment, the level of sediment at the time of the creation of the rock art has to be below the image. Unlike artefacts which are deposited alongside sediment, only sediment that accumulates after the engraving event will end up covering rock art images. If we were to rely on the method the reviewer suggests for dating the rock art of dating covering sediment, we would have an artificially short and young chronology for the art. We disagree with the view that a tool used to produce rock art cannot be used as evidence for the age of rock art - we are interested in dating the creation of the rock art rather than its burial, which as the reviewer says will have happened an unspecified amount of later.

The petroglyphs were necessarily made with stone, and since stone does not decay, we should expect to find some of these tools in spatial association with these large engravings. Our analyses match previous research by Hilbert et al. (2022), that has shown that the tools used to engrave sandstone are often left directly below the engraved panels. The shape of the tool and the battering marks show that it could not have been used for any other stone-working task. There is no likely scenario in which a tool that was used to peck sandstone could have ended up in a sealed archaeological context directly below an engraved panel but not been used to make this rock art. There is no other rock art within a distance of 200 m from the location of the pecking stone (all rock art on this boulder is phase 3 and 4 – shown in the new Figures 2C and 4). The only reasonable assumption in this case is that the tool was used to produce one of the two engravings above.

Our research focusses on the stratigraphic connection between the OSL samples and the engraving tool (pecking stone). Previous research at the Camel Site has also shown that large, naturalistic representations of Camels are Neolithic or older (Guagnin et al., 2021), based on results from OSL dating beneath fallen boulders and XRF analysis of rock varnish (Guagnin et al., 2021; see also Andrae et al., 2020; Macholdt et al., 2018;

Charloux et al., 2022). What is new about our research is that we can provide a more precise date by having discovered an engraving tool in a stratified context rather than lying on the surface, and that we have found engravings on a much larger scale – they are not limited to a confined area but spread along routes in the landscape.

Considering rock art, I noted that the camel shown in grey in figure 3 has its hump very far back. To me this does not look like a normal dromedary that one would expect as part of the local wild fauna. I was wondering if this could then be a Bactrian camel with two humps or an interbreed between dromedary and Bactrian (in Arabia called “bukht”)? Bactrian camels did not occur in Arabia naturally and were only imported during the Iron Age. The authors should provide additional information on the camel species and should consult an expert for camels. I further noted that the lower parts of the legs in figure 3 are shown with dashed lines. What does that indicate? Were the engravings incomplete here? This also needs to be clarified since this is important when assessing the spatial relationship between engraving, the sand deposits and the OSL sample locations.

We agree that the hump is very far back. We think the camel is actually shown in the process of getting up from a sitting position – and we do have some engravings of sitting camels in our dataset. The hindlegs of this camel have not been preserved, so we cannot prove this interpretation and have thus not included it in the manuscript. We also know of several “failed” engravings, where lines were placed in the wrong position, and corrected in a later engraving. In our previous paper we show a panel where the lines of a hump were corrected, and then superimposed with a new engraving of a camel that is apparently also in the process of standing up (Guagnin et al., 2023: 2B). We agree with the reviewer that the depiction of these details merit further analysis, but this is beyond the scope of this current manuscript.

Saudi Arabia has indeed got some very early representations of hybrid camels, but these animals were only introduced at the end of the 2nd or beginning of the 1st millennium BC. These engravings are generally associated with inscriptions, and follow the very different stylistic conventions of Iron Age rock art (they are small, only ca 20cm high, and lack detail – except the characteristic double hump). Crucially, the weathering and varnish build up preserved on some of the better preserved panels in our study has been shown to take at least 8000 years to form (Andreae et al., 2020; Macholdt et al., 2018). This difference is so pronounced that it is clearly distinguishable with the naked eye. It is therefore not possible that any of the representations show domesticated or “interbred” camels.

We have added extensive information and data on rock varnish throughout the manuscript:

- We have added a paragraph on rock varnish formation in the “Monumental rock art” section, as well as references to relevant research in Arabia and in the Sahara.
- We have also referenced figures that show this varnish in the manuscript and Supplementary Information.

- We have added a detailed photo of the engraved lines to the new Figure 4, which shows the engraving technique on panel ARN3 and also small areas where desert varnish has survived the extensive erosion evident on this panel.
- We have added a sentence on rock varnish in the discussion and referenced key publications.

The lower parts of the legs are shown with dashed lines to indicate they were covered in sediment. We have clarified this in the caption.

The archaeology and its age estimations on the other hand are comprehensible and well demonstrated. Although for the presentation of new archaeological material I miss a detailed description of how the excavations have been conducted. Did the authors sieve for example? How did they remove the sediment, all area at once or did they subdivide the excavated area as it would be a modern standard procedure? How did they document the position of the finds within the excavated layer? In Suppl. Fig. 11 there is no obvious instrument that would allow measuring the position of the findings, so how have the authors done the documentation? Moreover, what does for example "layer 5 (context 006)" as mentioned in the suppl mean? What is the context in this case?

We state in the methods section that trenches were excavated using the single context method. We now provide a reference for this method for clarification. We state that all sediment was sieved through a 3mm sieve and that trenches were hand dug using a trowel. Areas were not subdivided as our trenches were comparatively small at either 2x1m or 1x1m. The positions of finds were manually measured from an origin in one corner of each trench using a line-level and plumb bob. Excavations were carried out in May of 2023, with the use of technical equipment limited by the extreme heat and unnecessary in small-scale excavations. We have now included a description of our measurements in the methods section.

Our supplementary material noted the context number from excavations alongside the layer number with the intention of providing more clarity and easier cross-referencing with excavation notes. We realise this may have been unintentionally confusing to the reader. We have removed the context numbers throughout the SI document for clarity.

The counts on lithics provided, what do they include? How many lithic artifacts larger than 2 cm did the authors document and how is their horizontal distribution? Please avoid expressions such as "many" or the "majority" when presenting numerical data. Provide numbers instead to allow the reader to follow your presentation. So, while I consider the archaeology relevant, the presentation of the archaeological excavations and findings require significant revision following my questions above.

All lithics recovered during excavation are included in the analysis, we specify this in the methods. We do not use size cut-offs as this biases lithic analyses, particularly in the case of microlithic assemblages such as these. We have now added a figure (Supplementary Figure 14) showing the horizontal bedding of the lithics in ARN3 T2. We have deleted the

single use of the word majority from the main text and the three uses of the word many from the supplementary.

Beside my concerns regarding the age of the engravings, the other major issue is the scarcity of convincing evidence provided by the authors for the claim of “more humid conditions with increased water accumulation” (page 6 of the main manuscript) demonstrated by the “prolonged sequence of playa deposits” (ibid). Playa deposits is a flexible term that I think needs to be defined. Given table 1 and the grain sizes listed there, all but one context is composed of fine sand, which might be washed or blown in. In my understanding this is not the classic type of sediment that forms exclusively in standing water bodies. I think the reader would much appreciate it if the authors would define what they consider to be playa deposits exactly and how they differentiate between different depositional conditions.

Reading through their description of the palaeoenvironment results left me somewhat confused since there are contradictory statements. On the one hand they say: “[...] subordinate carbonate concentration, and the absence of root traces and organic-rich layers suggests that conditions remained too dry for the establishment of more permanent water bodies, indicating that a dryland environment persisted [...]” (page 6 of the main ms). While only a few sentences later they say: “These sites thus represent the earliest evidence from northern Arabia of increased humidity following the hyper arid LGM. However, the lakes were ephemeral, indicating an arid or semi-arid climate prevailed in the region.” (page 7, main ms). So what is the author’s final conclusion? To me the presented data clearly speaks in favor of a continuity of arid conditions with infrequent rain events that led to the formation of some shallow puddles, which have existed for a few days. A phenomenon that can also be observed today in Arabia and other deserts after rain. If there is no further supporting evidence for a longer duration of water bodies from these sections, such as the presence of pointer species for lacustrine conditions, these deposits can in my view not be cited as evidence for increased humidity.

We clearly state that these landscapes were arid with seasonal lakes (see Abstract) and that arid or semi-arid climate prevailed (see Regional Geomorphology and Palaeoenvironments). Conditions were more humid than the hyper arid conditions of the LGM – which is also the case for today’s climate. We also clearly state they were not humid enough for the establishment of more permanent water bodies – i.e. less humid than the Holocene humid period. We see no contradiction. Our intention was to highlight that this environment was indeed still extremely arid, with apparently just enough seasonal water to allow people to survive.

We have changed the wording from “increasingly humid” to “gradually more humid” to clarify we talk about an increase in humidity, not “humid conditions”. We have also added three sentences to show how these changes in climate would have changed the balance between erosion and sedimentation by changing from an environment dominated by aeolian erosion in the LGM to one subsequently dominated by fluvial sedimentation which often includes fine sand.

To wrap up, I think the presented conclusions for “securely dated rock art” and “the earliest evidence for increased humidity from northern Arabia” are not supported by the presented data. While saying this, the presented archaeological data is, although not comprehensively presented, at least convincing with regard to the chronology and cultural context of the population settled here at some time between 13.8 ka and 10.9 ka ago. That the authors can link this presence to Levantine Khiamian tradition is very interesting and would be worth further exploration. The excavations are relatively small since I suppose they have been started initially to collect chronometric data for the rock art. It might be worth going back, open larger trenches and see what the groups carrying Khiamian traditions else have left in the deep desert. The 500+ artifacts (if not 490 were smaller than 5 mm, please clarify the frequency of artifacts larger than 2 cm) could hint at an archaeological record of decent size. Finally I would like to point out that the dating of the Khiamian material from this site is relatively early compared to the standard time range proposed for the Levantine sites. Again an interesting detail that could be further explored.

We are pleased the reviewer is convinced by the age and characterisation of the occupation associated with the rock art. The radiocarbon date from ARN3 Trench 2 falls right within the Khiamian tradition, while the OSL date from Trench 1 and the OSL date from Jebel Misma 8 overlap in range with it. Other finds from these sites point to a subsequent PPNB occupation. We think there is strong reason to think these were the groups who made the rock art since we have recovered an engraving stone in a dated occupation layer and there are no other candidate early occupations at these sites.

We agree with the reviewer that these sites are worthy of further exploration, but our findings need to be published before we can seek further funding.

Reviewer #2 (Remarks to the Author):

In this article, the collaborating authors draw upon multiple strands of archaeological and paleoecological evidence to reconstruct a late Pleistocene-Early Holocene environment and populate it with fauna and human groups. For the latter, the archaeological evidence offers chronological and some cultural details such as technical affinities with northern traditions and long-distance exchanges of exotic materials. Mobility of these human groups is implicit, by virtue of the seasonal availability of local resources and potentially by depictions of seasonal characteristics of fauna. A major focus of this paper is the local rock art, studied in three newly reported locations and linked to well-reported rock art studies across a broader region of northern Arabia. While rock art is rich in imagery and clues to the symbolism and communications of its creators, it is rarely possible to date it or to link rock art unequivocally to other details of the archaeological record. The significance of this paper lies in the assertion that the earliest phase of this reported rock art can be securely assigned to the Late Pleistocene-Early Holocene transition and that the reported artifacts show an expansion of people into formerly hyper-arid zones after the Pleistocene LGM abandonment. This argument supposes that rock art of Phase 1 serves as a marker of human activity earlier than previously documented Late Pleistocene-Early Holocene Arabia.

There are two key strands to this argument for which the authors supply details and which this reviewer is best qualified to assess. (Paleoecological details of sediment characteristics and dating methods underpinning the regional environmental reconstructions, including paleolacs, are not my forte). One strand is the age of the rock art and its indirect dating through purported association with artifacts. The second strand is the absence of evidence for Late Pleistocene and Epipaleolithic occupation and use of local resources to purport a re-population/expansion, driven by exogenous climatic factors) into a region previously lacking human occupation. This second strand is the major justification for publication in Nature Communications, even as the evidence and topic of rock art is intrinsically interesting, if localized. Similar and intrinsically localized evidence can be of the highest scientific interest, as studies of 40,000-year-old cave paintings in Southwest Sulawesi have demonstrated.

We thank the reviewer for their succinct summary of our paper and for highlighting the importance and intrinsic interest of rock art research. We would like to clarify that the absence of occupation is documented between the LGM and ~14ka; possible evidence of Epipalaeolithic occupation dating to 13.4 ka was documented at Sahout (Guagnin et al., 2023). We have added a sentence and two references in the first paragraph of the discussion to clarify the timing and nature of this gap in occupation.

To examine these arguments, therefore, some close consideration of the linkages and chronologies of rock art and excavations is warranted. While the evidence is well presented, very well illustrated, clearly organized, and amply supplemented with extensive online supplemental materials, there remain some ambiguities. It is fortuitous to find sediments sealing or obscuring rock art as was the case at site ARN3 T1. This is the only fully reliable stratigraphic dating (indirect) for rock art.

The engraving stone must have been used to engrave one of the two camels above, as there is no earlier (or later) rock art within a distance of 200 m from this site. Based on the associated OSL date we assign these camels to the PPN (phase 3 and 4) of the rock art.

A critical item therefore is the white camel (phase 1) on the panel ARN 3 with its legs covered in sediment. The authors assign this to the Pre-Pottery Neolithic (layer 5 in ARN T1). With a rockfall (layer 4) overlying layer 5, the camel engraving (and all phase 1 rock art classed as contemporaneous with it by the authors) could be later than layer 5, engraved independently and sealed at a later date by rockfall. While the evidence is strongly suggestive of a PPNA (not Natufian) association with layer 5 and therefore of the rock art "engraving" of a camel, questions remain. The tool argued to be used for engraving is also described as a "pecking stone" (Supplement 3.3 p. 16-170, and this pecking motion is also described in Supplement 3.4 p. 42 (Figure 37 on p. 44). I would like to see clarification of

- A) The reasons for dating to Natufian (a few notched flakes and mortars) instead of PPNA (later, consistent with and unremarkable for other finds in Arabia, e.g., Jebel Oraf). Note another "pecking" tool recovered on the surface at ARN30 (rock

art production method not specified in Supplementary Table 1). Radiocarbon dates are consistent with PPNA, not Natufian—unless one reads the most ancient extent of the radiocarbon determination rather than its median.

This is a mis-understanding, we are claiming that the main occupation at Jebel Arnaan (and the earlier phase of camel engraving) was PPNA not Natufian. We have reworded to clarify. We state that occasional typologically Natufian artefacts were recovered, but we do not use these artefacts to date the rock art to the Natufian. There is earlier rock art than the camels featuring human figures that could relate to the Natufian, but this is not associated with datable archaeological contexts and therefore not the subject of this paper. We have removed three of the mentions of the Natufian from the paper to make it clearer that we think that large camel tradition belongs to the PPN.

B) The difference between engraving and pecking and the reasoning for a pecking tool to be an engraving instrument. This is critical for the association of the tool in ARN3 T1 and the camel engraving above it. I suggest more text on how the rock art was produced (pecking, engraving, other?) and explanation whether production criteria was included in the analysis of phases/chronology—or do different production techniques cross-cut the chronological assignments of phases? I also suggest (in light of the access challenges) that authors address whether one or many people were needed and consider the person hours of labor involved. These considerations may lead to further insights on group size, frequency of passage, and community dynamics.

The engraving process can consist of 1 or 2 steps. First lines are pecked. This method is used in all known panels. Some images are “scraped” in a second step, which turns the rougher pecked lines into smooth grooves. We have found both types of tools, but the one described from the excavations in ARN3 T1 is a pecking rather than a “grooving” tool.

We have clarified this throughout the manuscript:

- We have included a description of engraving process in the “Archaeological excavation” section: “Analysis of the engraved lines shows that all images were pecked using a pecking tool. In some engravings, pecked lines were smoothed in a second step, using a smoothing tool (**Error! Reference source not found.** inset).”
- We have included an inset in the new Figure 4 that shows a close-up of the pecked/engraved lines on ARN3.
- We have included a sentence in the “Artefact assemblage” section that explains the tool was used to peck rather than smooth lines.
- We have changed the wording in the caption of Figure 6 to “pecking tool” to clarify this distinction.

We agree with the reviewer that it would be interesting to quantify the effort involved in engraving these lines, but this would require a dedicated experimental study which is beyond the scope of this paper.

- C) The reason the camel engraving cannot be later than layer 5, that is, PPNA occupants at layer 5. As I read their evidence, it is plausible that there is an abandonment layer, then layer 4, which covers the camel engraving of rock art phase 1). It is possible that the camel engraving postdates layer 5 (by an unspecified period of time) and pre-dates layer 4 collapse (documented by authors in section Figure). It would seem that a post PPNA composition of phase 1 would be very much consistent with evidence elsewhere of PPNB cultural influences among populations in the north Arabian humid period.

If we assume that the camel engraving post-dates layer 5, then we have a pecking stone that shows signs of having been used to engrave sandstone, but no engraving to go with it. There are no earlier engravings below the two camels. We therefore assume that one of the camels above was produced with this tool.

Our observations also show that engravings are normally placed higher up and not directly above the ground presumably to make pecking easier and protect the engravers' hands. The ground surface would have been close to the camel's feet during the deposition of layer 4, making engraving difficult. While there does appear to have been PPNB occupation at Jebel Arnaan, the main occupation layers in both trenches date to the PPNA. We assume rock art phase 3 begins in the PPNA, but large animal representations appear to have been produced over an extremely long period and well into the PPNB.

Note that if the evidence is stronger (as I read it to be) for PPNA not Natufian incursions into the arid zone, the chronological assertions are less remarkable than the authors indicate, or they could be re-stated for local versus peninsular significance. (See Crassard et al. 2013, cf., Hilbert et al. 2014 both cited by authors). Even so, the analysis classifying different episodes of rock art based on stylistic grounds and overlap/superposition of images is new and potentially foundational in the region.

The reviewer is correct, our present paper has stronger evidence for PPNA incursions. We took the evidence for possible Natufian incursions from previous research at SAU, where we have an even earlier radiocarbon age. We have re-worded this in the conclusion and also referenced where the information comes from. We have reduced the claims for a Natufian presence although some of the artefacts (the Helwan bladelet and the opposed notch blades) and some of the dates do still suggest this. We have clarified our argument that the life-sized naturalistic camel engraving tradition belongs to the PPN, not the Natufian.

The second strand is the absence of evidence for Late Pleistocene and Epipaleolithic occupation (at this locale) and use of local resources to purport a re-population/expansion, driven by exogenous climatic factors. The authors need to

broaden the significance of this paper beyond the repopulation of this specific locale in the Arabian Peninsula. This might be done by referencing other and adjacent geographies: the greening of the Sahara or the LGM refugium in the Arabian/Persian Gulf theory come to mind.

Supplementary Figure 31 shows the only older diagnostic artefact than the final Epipalaeolithic which was found during a fieldwork: a Middle Palaeolithic Levallois core. We have added a reference to the site of Al Marrat 3, the youngest Middle Palaeolithic site in northern Arabia, with no known intervening occupations between this and the sites documented in our project anywhere in northern Arabia.

A few minor points:

Caption in Figure 1 (main text) documents 131 rock art images, versus (in text) account of 176 images (line 150).

Text and caption both list 130 life-sized images (the text additionally lists "2 camel footprints, 15 smaller 153 scale naturalistic depictions of camels, 19 human figures, 4 human faces or masks, and 6 154 unidentified, partial engravings". We have now included these in the caption.

Line 234—dating the engraving of the camel by the height of its legs. Even if the authors' reasoning is sound (? See above), this line is confusing and needs to be clearer for first reading.

We have changed the wording in this sentence to "the fact that the legs of the engraved camels were covered with sand".

Line 297 and ff. lithics—Supplement p. 36 & Fig 20—that there is a transition in materials between PPNA and PPNB layers is interesting and deserves greater comment. A gap in time? Different populations? Different economic pathways for the circulation of goods?

We agree with the reviewer that this is an interesting aspect of the archaeology worthy of greater comment, but it is beyond the scope of this paper. A follow-up paper in preparation focusses more on the lithics, including the silcrete source at Sahout and geochemical provenancing of the obsidian. In this next paper we describe the mobility implications of the differences in material between the PPNA and PPNB and we discuss the issue of whether there is chronological hiatus using new dating evidence from Sahout.

Line 379 argues that engravings reflect a rainy season. I tend more to agree with the multiple and sometime symbolic/ideological and/or social communications the authors explore in discussion. It is altogether too easy to "read" as economic and functional-historical images one doesn't access through cultural practice and initiation. I'd prefer to keep the interpretation open and refrain from linking specifically to moisture/rainy season.

We have removed this line.

Supplementary Figure 10 has a dark shadow that obscures critical section information. The authors would be forgiven for photoshop manipulation of this figure to illuminate any section details that can be discerned.

We have removed the shadow as much as possible (using Adobe lightroom).

Likewise it is odd that the authors elected not to excavate fully adjacent/perpendicular to the limestone face bearing the rock art (as they did in ARN3 T2). If this was done subsequent to the image, a description of the lowest extent of phase 1 camel engraving (outlined in black) would be warranted.

We initially left a baulk between the trench and the engraving to protect the engraving during excavation and to get a better view of the stratigraphy. This was later removed (see Supplementary Figure 11). The lowest extent of the camels (which are phase 3) is shown in the tracing provided in Figure 3, although the legs and feet of the earlier camel are partially eroded and no longer visible.

We have clarified with a note on the erosion of camel legs in the caption of Figure 4; information on the bulk and its removal in the caption of Supplementary Figure 10; and we point out the removed baulk in the caption of Supplementary Figure 11.

Missing references include other efforts at dating rock art in Arabia, either through indirect association with sediments (Rachad and Inizan's work at Sa'ada, Yemen) or through stylistic classifications and their superposition in different panels (also used in this manuscript) (e.g., Philby-Ryckmans' expedition publications). Direct dating has recently been possible with (organic) painting (Rowe et al. The Holocene 2022).

We have deliberately not referenced the superpositions described by the Philby-Ryckmans' expedition (photos of which were analysed by Anati), as these descriptions are now considered controversial by some researchers (Bednarik and Khan have written several critiques).

We have included a short section on rock art dating based on varnish analysis and reference work by (Andreae et al., 2020; Macholdt et al., 2018) that is specific to petroglyphs in Saudi Arabia in the "Monumental Rock Art" section. We have referenced figures in the manuscript and SI that show rock varnish in our dataset, and we have included a close-up of pecked lines with patches of desert varnish in the new Figure 4. We have also included a section explaining the context of rock varnish analyses in the Discussion.

Reviewer #3 (Remarks to the Author):

A very interesting piece of work. It is good to see the archaeology tied to the rock art, together with the various dating methods. A few remarks:

Line 93 What stone are the carvings on (I assume sandstone, but please state at the outset)

We have changed the wording to “sandstone outcrops”.

Line 160 (Fig. 1) A: Map could be larger and B possibly two gazelle and one goat

A: We have increased the size of the map in a new standalone map Figure 1.

B: Comparison with other panels that also show the bodies of gazelles suggests all three engravings also represent gazelles. Goats are now known in the rock art until the Neolithic, where they are generally shown with horns that curve upward in parallel and then outward at the top, as well as with much more heavily set bodies (Stewart et al., 2024: Fig 6).

Line 205 Freshly engraved INSERT COMMA the images...

Comma inserted.

Line 366 omit comma

Comma deleted.

Line 374 could the aurochs not be a record of what has been seen elsewhere?

That is exactly what we meant when we said “...or that during dry seasons human populations retreated to wetter areas where aurochs were present”. For clarity, we have changed the sentence to “...or that aurochs were seen elsewhere, perhaps when human populations retreated to wetter areas where aurochs were present”.

In the paragraph starting on l. 431 maybe add something about representation of the animals also being a record of the environment, potentially sympathetic magic, territorial markers, indications of herd movements (an extension of the statement about routes l. 437),

We have re-worded the paragraph to include territorial markers. We would prefer not to speculate on magic, as this draws parallels to much more recent customs in South African rock art and we have no evidence for this interpretation.

Line 489 the monumental rock art in Arabia sets them apart from the Levantines—maybe a phrase to explain what sort of rock art was found in most of the Levant... Also, maybe worth noting that such monumental rock art is found in North Africa, though not in Egypt. And, it might be worthwhile speculating (if possible) why monumental images were so common here.

We would like to avoid comparisons with the Levant and the Sahara at this point. Such a comparison warrants a much more detailed article. Part of the problem is also that to date only very little rock art that pre-dates the Iron Age is known from the Levant. We

are aware that one key site of Neolithic depictions (small cattle engravings) has recently been discovered in Jordan but has not yet been published. At present this prevents any meaningful comparison between both regions.

What animal bones were recovered from the area? Can these be linked to the rock art?

We provide a report on the faunal remains in Supplementary Note 4. Sadly, the material was so badly preserved that no species identifications were possible, with exception of a piece of ostrich eggshell.

Reviewer #4 (Remarks to the Author):

The manuscript entitled: "Monumental rock art illustrates that humans thrived in the Arabian Desert during the Pleistocene-Holocene transition" presents highly original data from a key archaeological site in Northern Arabia. With regard to the interdisciplinary scope and the relevance of the topic presented in the Ms I have no doubt that this manuscript is a potential very good fit for Nature Communications as it is relevant for the broader scientific community. Also, with regard to the highly original multi-method dataset this Ms deserves publication in a high-ranking outlet. However, I was a bit less impressed with the structure and conciseness of the main manuscript. The manuscript is occasionally redundant and the argumentation is – at least partly – not convincingly structured. Even more importantly it is oftentimes difficult to trace back the scientific evidence that should support the argument/interpretation. Also, some of the figures need to be improved especially with regard to their self-explanatory power. From my perspective it is crucially important that the actual scientific evidence is directly associable with the corresponding figure (especially Fig. 6) or interpretation. In summary, I see the high potential of the paper and I am convinced that it can have a significant scientific impact, however, a major revision is required to make it fit for publication. Below I provide some feedback that will hopefully help to make the Ms more concise. With regard to the supplement, I focused on the geomorphological (note 5) and the luminescence data (note 6) and provide some additional feedback especially with regard to documentation and interpretation.

We thank the reviewer for highlighting the relevance and importance of our manuscript.

We have improved the structure of the manuscript to make our argument more concise:

- We have re-written the paragraph introducing our research.
- We clarify how rock varnish indicates the age of engravings and referenced recent research.
- We have linked figures better with evidence from scientific dating and now include dating information in figures 3, 4 and 7.
- In the discussion we have made our statement on potential aurochs movements more concise, and we have clarified the link between the excavated pecking tool and engraved rock art.
- We have shortened the conclusion to provide better focus on our results, while providing additional references for comparative information.

We have changed several figures to make them more self-explanatory (for more detail see responses to comments below):

- We now provide a stand-alone map in figure 1 that gives a better overview of the sites and landscape.
- We provide an extended selection of engravings in the new figure 2.
- We provide additional information on sediments and engraving in the section drawing in figure 4.
- We have provided more extensive information in the caption of figure 7 (formerly figure 6).
- We have provided clearer maps in figure 8.

Major comments:

- 1) Introduction line 74ff: I do not find it helpful that the authors provide a summary already in the introduction, which will ultimately cause redundancy. Maybe it's a matter of personal taste but to me this para reads a bit like a cheap sales-pitch. I'd recommend to use this space to more convincingly carve out the goal setting of the paper and maybe establish some main hypothesis here.

We have re-written this paragraph to state our initial hypotheses rather than pre-empt results.

- 2) Figure 1, line 159f: also, I'd recommend to redo this figure, especially sub-figure A. The informative value of the maps shown in supplementary figure 40 is much higher and they also meet scientific standards with regard to north arrow and a proper scale. Throughout the main manuscript I found it difficult to link the archaeological as well as sedimentological findings to the geographical/geomorphological context. A better map shown here could be of tremendous help in that regard. Regarding sub-figure B to D I think a readable scale is missing. I know there is one in the photo, however, it is almost impossible to read and to use as a proper scale.

We have increased the size of the map and turned it into a standalone figure (new Figure 1). We have also included a north arrow.

We have made a new Figure 2 with the rock art images. We now clearly indicate the size of the scale in the caption.

- 3) Sampling context (e.g. Fig. 2 but also more general): It is difficult for readers who have not been part of the excavations to link the archaeological excavations to the sampling context for sedimentological and geochronological analyses. For example, in Fig. 2 it would be nice to get a direct impression where the sampling site for OSL and radiocarbon dating is located and what the connection between the engravings and the geological record is. This link needs to be easily traceable in the main manuscript as it provides a main line of evidence for the later interpretation e.g. presented in Fig. 6 and the discussion section.

We have included the location of the trenches in what is now Figure 3. We have also included a description of the playa sampling location and a reference to Figure 8 which shows them on a satellite image. The new Figure 1 with the larger map should also make these correlations clearer.

- 4) Interpretation of playa sedimentation (e.g. line 360, or supplement 5 section 5.3.): the authors use playa sedimentation as a local paleoenvironmental records especially with regard to water availability, which generally is a good idea. However, the authors ignore in their analyses and interpretation of the record that wind erosion could have played a significant role in forming this playa sediment successions, which would imply that the claim of the first humid phase since the LGM is a bit on shaky ground. Alternatively, one could interpret that a previous humid phase is missing in the record either because there was not enough water available in the catchment to form playa sediments prior to this phase or the corresponding playa sediments had been subsequently eroded by wind erosion during a dryer phase. From my perspective are both scenarios possible with an almost equal probability.

The reviewer has raised a good point here that we did not articulate in the paper. We have now added three sentences that address this issue by discussing how the balance between aeolian erosion changed from the LGM to the Holocene:

“We interpret the onset of playa sediment accretion as representing the change in the balance between aeolian erosion and fluvial sedimentation. During the LGM the hyper arid environment meant that any fine-grained sediments deposited in ephemeral floods were subsequently eroded by the wind. As the climate became less arid, fluvial sedimentation increased, becoming greater than the aeolian erosion rate and sediments started to accumulate.”

- 5) Interpretation of the results (e.g. Fig. 6): as addressed already in my major comment#3 it is very difficult trace back the actual evidence that form the basis for the interpretations presented in this section and figure 6. As this is a scientific paper and journal, I think that the connection between the argument and the underlying scientific evidence should be as strong as possible. That said, I recommend to majorly revise Fig. 6 with the goal to make all links to the actual scientific evidence clearly visible. For example, the interpretation of the environmental context in the very right column, what is it actually based upon? The evidence needs to be directly traceable. Same with the columns human activity and rock art.

As stated in the caption, all information provided in black font reflects results and interpretation provided in the text. We have added more detailed statements of what evidence each set of data is based on in the caption to provide more clarity. The key pieces of human activity and rock art evidence are described on the figure itself. For the environmental evidence in order to keep the figure easy to read it is best to provide the basis for the interpretation in the caption rather than in the figure itself.

The climate interpretation in figure 6 is based on our interpretation of the sedimentology and chronology as discussed in the text.

- 6) Supplement 5, Figure 42: the header “grainsize and texture” is a bit to vague. What statistical parameter did you use to describe the underlying distribution of grainsizes? Is it D50 or something else?

Grain size and sorting was observed visually and textually in the field and classified according to standard the sedimentological classification. This information has now been added to Supplementary Note 5.

- 7) Supplement 5, Figure 42 and page 56, section 5.3: why was the sandy gravel layer only sampled for dating in JMI and not in ARN? With only one age on this unit the interpretation that it belongs to very late MIS5 (page 56, section 5.3 first para of this section) is shaky and I think this should be highlighted in the text.

We attempted to sample the gravel. However, it was very coarse and the OSL tube would not penetrate. Furthermore, a block sample also failed as the sediments were not cemented. The late MIS5 age is consistent with other evidence for humidity in northern Arabia during this period which we cite in supplementary references 37-40.

- 8) Supplement 6: to show the general robustness of both the multiple-grain as well as the single-grain results the authors need to show dose recovery results, which is the standard performance test. Without dose recovery results the luminescence data is difficult to trust.

Dose recovery tests were performed on five of the eight samples presented in this study, with all five samples yielding acceptable (within 2σ of unity) ratios. This information has been added to Section 6.1.4 of the Supplementary Notes.

- 9) Supplement 6, section 6.1.5: to be able to judge on the appropriateness of the interpretation of the De distributions and the application of the different age models to obtain the burial dose I strongly recommend to provide De distributions (preferably as radial plots but KDE plots would also work) in supplement 6. Without De distribution is difficult to judge on the robustness of the obtained Db and thus OSL ages.

These plots have been added as Supplementary Figure 45.

- 10) Supplement 6, section 6.1.5: The use of the FMM for sample JMI8-T1-4 seems problematic to me in this context.

- a. To me the more appropriate age model for this sample would be the MAM especially when using the bootstrap version (Cunningham and Wallinga 2012) where σ_b can be assigned with an uncertainty (e.g. $25 \pm 5\%$).
- b. The FMM would only make sense if the sample is “contaminated” with young dose grains, something the authors recognize themselves (section 6.1.5, first para, last sentences), as the MAM would have difficulties to handle these lowdose outliers/population. However, in the next paragraph of section 6.1.5 you write that sample JMI8-T1-4 showed

high-dose grains presumably resulting from “contamination” from older and not younger grains. So, there is no added value in applying the FMM here.

- c. The way σ_b is determined by basically doing a sensitivity analysis using the FMM is not stringent if you ask me. The σ_b describes the unexplained scatter in the D_e distribution for a perfectly bleached and un-mixed sample. As you have samples that seem to behave this way, I strongly recommend to apply single-grain measurements to one of your well-behaving samples and use the obtained over-dispersion as input σ_b for D_b determination of sample JMI8-T1-4. If you apply the bootstrap MAM (and I think you should) you can assign an uncertainty to this σ_b (e.g. 20 % relative uncertainty), which is more realistic than using σ_b without any uncertainty.

In relation to points a and b, the MAM is very susceptible to being biased towards individual low values yielding inappropriately young ages (e.g. Rodnight et al., 2006). The bootstrapped MAM is an attempt to circumvent this failing, but it requires an appropriate σ_b value (see above) and remains sensitive to anomalously young grains within a small dataset. Owing to the very low yield of acceptable grains from JMI8-4, and the limited amount of material available from MIS-B-3, both datasets are small. Furthermore, JMI8-4 contains 1 anomalously young grain, meaning that Reviewer 4’s point b does not apply (though they couldn’t have known that without the dose distribution diagrams for the single-grain datasets, which they correctly asked us to add to the paper).

In relation to point c, there are two main objections. Firstly, the σ_b determination procedure used here is entirely standard and has been routinely used in the literature for nearly two decades (e.g. Jacobs et al., 2008, Section 9.1); it essentially balances improvement of the fit to the data (maximum log likelihood) with an assessment of the minimum number of components required to explain the data (Bayesian information criterion). This could be characterised as a “sensitivity analysis”, but it is still a sensible approach to fitting the data. Secondly, the samples have a low intrinsic luminescence sensitivity and low a low equivalent dose, meaning that the yield of measurable grains is $\sim 0.002\%$. Assuming that well-bleached samples from the study area are a good analogue for JMI8-T1-4 (not necessarily a given), generation of a large enough dataset to precisely constrain σ_b would be prohibitively time-consuming. Given that a widely used and well accepted approach to determining σ_b from the JMI8-T1-4 dataset itself exists, we have preferred to use that method.

Some additional minor comments:

- 11) Line 196ff: Please rephrase these two sentences starting with “The engravers....”. There are non-scientific and speculative.

These observations are based on experiments and on our experience in the field. We have changed the wording and now start the sentence with: “The friable nature of the substrate and the slope of the narrow ledge suggest...”. We have rephrased to say

they 'likely risked their lives' rather than they 'must have', but the risk is clear and already stated in the previous sentence as to why these panels were recorded with a drone.

12) Line 204ff: also, here I recommend to tone down the overly sensational tone of this paragraph.

We have rewritten the first sentence of this paragraph and removed the speculation to 'even millennia'.

13) Figure 3: please provide the ages in the figure to make it easier to link engravings to the geochronological framework (also see my main comment#3). Also provide some more details on the kind of sand (e.g. with regard to texture, bedding structures) deposited here.

We have amended the figure to include the OSL ages and details on the sand directly in the section drawing.

14) Line 400: "...pre-date the tradition of life-sized camel engravings..." → which is evidenced by?

They must be older because they were found underneath later camel engravings. We have changed the wording to make this clearer: "...stylised human figures (rock art phase 2, **Error! Reference source not found.**D) were repeatedly found below engravings of life-sized camels (phase 3). These human figures consequently pre-date engravings such as those at ARN3, and are thus likely older than 12 ka (**Error! Reference source not found.**)".

15) Figure 7: where is North?

We follow the convention of having north at the top of all maps. We have now included north arrows to avoid any confusion.

16) Supplement 6, page 59, section 6.1.4: please state size of the multiple-grain aliquots
This information has been added to the first line of section 6.1.4.

17) Supplement 6, page 59, section 6.1.4: indicate in table 8 which ages are based on multiplegrain and which ages are based on single-grain measurements to make it easier to follow. That said, I'd also add the information on the age model applied to the table.

This information has been added to the caption of Supplementary Table 8.

18) Supplement 6, page 59, section 6.1.4: why was an exponential plus linear fit of the dose response curve was used? There is no reason based on physics to use this.

True, but there is a practical reason, which is that the exponential plus linear fit is often the best fit to the measured data. In principle, dose response curves should be best described by a single saturating exponential (the single-trap model) or by the sum of two or more saturating exponentials (two or more traps). In practice, the single exponential fit does not always adequately describe measured dose response curves, and curve fitting software does not invariably produce convincing fits using two or more saturating exponentials, so the saturating exponential plus linear fit is a pragmatic (and

widely adopted) solution. At large equivalent doses (>~100 Gy), use of different fits can have a statistically significant effect on the calculated equivalent dose, and careful justification of the model used is necessary. Conversely, in the present study, all of the ages supporting the archaeological argument are derived from samples with equivalent doses <40 Gy, in which case most of the dose response curve shape is derived from the saturating exponential component of the fit i.e. the difference between a saturating exponential fit with or without a linear component is negligible. Consequently, we have retained the exponential plus linear fitting used in the previous iteration of the manuscript.

References cited in the response

- Andreae, M. O., Al-Amri, A. M., Andreae, C. M., Guagnin, M., Haug, G., Jochum, K. P., Stoll, B. & Weis, U. 2020. Archaeometric studies on petroglyphs and rock varnish at Kilwa and Sakaka, northern Saudi Arabia. *Arabian archaeology and epigraphy*, 31, 219-244. <http://dx.doi.org/10.1111/aae.12167>
- Charloux, G., Guagnin, M., Alsharekh, A. & Petraglia, M. D. 2022. A Rock Art Tradition of Life-sized Naturalistic Engravings of Camels in Northern Arabia: New Insights on the Mobility of Neolithic populations in the Nafud Desert. *Antiquity Project Gallery*, 96, 1301-1309. <https://doi.org/10.15184/aqy.2022.95>
- Guagnin, M., Charloux, G., Alsharekh, A. M., Crassard, R., Hilbert, Y. H., Andreae, M. O., Preusser, F., Dubois, F., Burgos, F., Flohr, P., Stewart, M., Mora, P., Alqaeed, A. & Alali, Y. 2021. Life-sized Neolithic camel sculptures in Arabia: A scientific assessment of the craftsmanship and age of the Camel Site reliefs. *Journal of Archaeological Science: Reports*, 103165. <https://doi.org/10.1016/j.jasrep.2021.103165>
- Guagnin, M., Shipton, C., Stileman, F., Jibreen, F., Alsulaimi, M., Breeze, P. S., Stewart, M., Hatton, A., Drake, N. A., Jha, D. K., Al-Tamimi, F., Al-Shamry, M., Al-Shammari, M., Kay, A., Groucutt, H. S., Alsharekh, A. M. & Petraglia, M. D. 2023. Before the Holocene humid period: Life-sized camel engravings and early occupations on the southern edge of the Nefud Desert. *Archaeological Research in Asia*, 36, 100483. <https://doi.org/10.1016/j.ara.2023.100483>
- Hilbert, Y. H., Clemente-Conte, I., Crassard, R., Charloux, G., Guagnin, M. & Alsharekh, A. M. 2022. Traceological analysis of lithics from the Camel Site, al-Jawf, Saudi Arabia: an experimental approach to identifying mineral processing activities using silcrete tools. *Archaeological and Anthropological Sciences*, 14, 93. 10.1007/s12520-022-01559-6
- Jacobs, Z., Wintle, A. G., Duller, G. a. T., Roberts, R. G. & Wadley, L. 2008. New ages for the post-Howiesons Poort, late and final Middle Stone Age at Sibudu, South Africa. *Journal of Archaeological Science*, 35, 1790-1807. <https://doi.org/10.1016/j.jas.2007.11.028>
- Macholdt, D. S., Al-Amri, A. M., Tuffaha, H. T., Jochum, K. P. & Andreae, M. O. 2018. Growth of desert varnish on petroglyphs from Jubbah and Shuwaymis, Ha'il region, Saudi Arabia. *The Holocene*. <https://doi.org/10.1177/0959683618777075>
- Rose, J. I. 2022. Seeking Solace (50–12 Ka). In: Rose, J. I. (ed.) *An Introduction to Human Prehistory in Arabia: The Lost World of the Southern Crescent*. Cham: Springer International Publishing, 231-255. 10.1007/978-3-030-95667-7_10

Stewart, M., Andrieux, E., Blinkhorn, J., Guagnin, M., Fernandes, R., Vanwezer, N., Hatton, A., Alqahtani, M., Zalmout, I., Clark-Wilson, R., Al-Mufarreh, Y. S. A., Al-Shanti, M., Zahrani, B., Al Omari, A., Al-Jibreen, F., Alsharekh, A. M., Scerri, E. M. L., Boivin, N., Petraglia, M. D. & Groucutt, H. S. 2024. First evidence for human occupation of a lava tube in Arabia: The archaeology of Umm Jirsan Cave and its surroundings, northern Saudi Arabia. *PLOS ONE*, 19, e0299292. [10.1371/journal.pone.0299292](https://doi.org/10.1371/journal.pone.0299292)

Response to reviewers

We thank the reviewers for the time they have taken to review our manuscript a second time. We are pleased all issues are now resolved. The reviewers' comments have improved the quality and readability of our manuscript.

Reviewer #1 (Remarks to the Author):

I thank the authors for considering all my suggestions. The paper is now in a state where I can recommend publishing it.

We are glad all issues are resolved.

Reviewer #2 (Remarks to the Author):

This is my second review, and I am grateful for an opportunity to see the changes made to the manuscript as well as the issues raised and responses to other reviewers. I consider this process akin to joining a review panel or scholarly jury after consideration and review of the work by myself. This comment is therefore influenced by my readings of other reviewer-peers and of course by my own re-consideration of manuscript and revisions. I caution that I have provided a more cursory reading than on first review. It is interesting to see that reviewers have raised some of the same concerns and points while bringing different expertises to this review.

I find that the authors have made substantive, highly informed and generally persuasive responses to the many detailed comments we reviewers have raised. There is no doubt that this work is of highest quality, that the methods and analyses are sound, cautious, and address significant questions. Notably, the repopulation and adaptations of human groups to arid environments has significance beyond the data set represented by this small cluster of sites in Northern Arabia.

We thank the reviewer for their positive assessment of our revised manuscript.

My lingering concern is the regrettable paucity of evidence that links the first phase (camel images) of petroglyphs to PPNA (the authors have corrected prior text that led me to suppose they argued a Natufian date). The interpretation still hangs on one pecking/engraving stone found 72 cm away near the top of a layer dated by OSL and by a hearth (14C) in a different trench. The excavation is small; as much as I sympathize with the authors' urgency to publish before seeking additional funding to expand the data set (their letter of reply), I worry that their interpretation still has privileges to an earlier date over potential explanation for a later one. 200 m is not so very far to transport a pecking tool (from a later engraving phase), possibly with the intent to use it, but instead it was lost or abandoned. It is my experience that sedimentation in the desert is neither continuous nor solely depositional. Where material accumulates (layer 5), it can also blow away, leaving heavier stone on the surface (desert reg). This can happen multiple times; thus what appears to be the upper part of layer 5 can reflect an accumulation of artifacts of different

ages and sources. Again, the evidence would be more compelling if the hearth was nearer the camel, if the camel legs were clearly complete, if there were more pecking stones or it sat in a dated hearth, or if the trench were extended to join the trench where the dated hearth occurs.

The reviewer is correct that sediment in the desert often blows away. However, in the case of the ARN T2 excavations refits from layers 8 and 9 attest to the high integrity of the archaeological context.

We have included a caveat in our discussion: “Nevertheless, the possibility that the engraving tool was transported from a different panel at Jebel Arnaan, cannot be excluded entirely.”

Archaeologists cannot always obtain the richness of data they would prefer—discovery relies on chance and expertise. My point is not to discredit the excellent work the authors have done, and their high expertise is evident.

We thank the reviewer for their positive assessment of our work. We agree that it is not always possible to obtain the richness of data we would all prefer.

Reviewer #3 (Remarks to the Author):

The revised work reads much better and address all the reviewers' concerns.

I agree with the authors that the artefacts for creating the images would be beneath (maybe well beneath) the images, though it is always tricky with rock art to always link archaeology to particular works.

It is hoped that more funding is forthcoming so that the authors can return to the site and extend the area of excavation.

We thank the reviewer for their assessment and are pleased all concerns are resolved. We agree that more funding and further excavations would help our understanding of these early sites.

Reviewer #4 (Remarks to the Author):

I would like to thank the authors for their detailed rebuttal. As a result of I think that the revised manuscript is much improved and that my main concerns are sufficiently addressed. I am happy to recommend the Ms for publication in Nat.Comm.

We thank the reviewer for their positive feedback. We are happy all concerns are now addressed.

Monumental rock art illustrates that humans thrived in the Arabian Desert during the Pleistocene-Holocene transition

Submitted to NatureCommunications

The manuscript entitled: “Monumental rock art illustrates that humans thrived in the Arabian Desert during the Pleistocene-Holocene transition” presents highly original data from a key archaeological site in Northern Arabia. With regard to the interdisciplinary scope and the relevance of the topic presented in the Ms I have no doubt that this manuscript is a potential very good fit for Nature Communications as it is relevant for the broader scientific community. Also, with regard to the highly original multi-method dataset this Ms deserves publication in a high-ranking outlet. However, I was a bit less impressed with the structure and conciseness of the main manuscript. The manuscript is occasionally redundant and the argumentation is – at least partly – not convincingly structured. Even more importantly it is oftentimes difficult to trace back the scientific evidence that should support the argument/interpretation. Also, some of the figures need to be improved especially with regard to their self-explanatory power. From my perspective it is crucially important that the actual scientific evidence is directly associable with the corresponding figure (especially Fig. 6) or interpretation. In summary, I see the high potential of the paper and I am convinced that it can have a significant scientific impact, however, a major revision is required to make it fit for publication. Below I provide some feedback that will hopefully help to make the Ms more concise. With regard to the supplement, I focused on the geomorphological (note 5) and the luminescence data (note 6) and provide some additional feedback especially with regard to documentation and interpretation.

Major comments:

- 1) Introduction line 74ff: I do not find it helpful that the authors provide a summary already in the introduction, which will ultimately cause redundancy. Maybe it's a matter of personal taste but to me this para reads a bit like a cheap sales-pitch. I'd recommend to use this space to more convincingly carve out the goal setting of the paper and maybe establish some main hypothesis here.
- 2) Figure 1, line 159f: also, I'd recommend to redo this figure, especially sub-figure A. The informative value of the maps shown in supplementary figure 40 is much higher and they also meet scientific standards with regard to north arrow and a proper scale. Throughout the main manuscript I found it difficult to link the archaeological as well as sedimentological findings to the geographical/geomorphological context. A better map shown here could be of tremendous help in that regard. Regarding sub-figure B to D I think a readable scale is missing. I know there is one in the photo, however, it is almost impossible to read and to use as a proper scale.
- 3) Sampling context (e.g. Fig. 2 but also more general): It is difficult for readers who have not been part of the excavations to link the archaeological excavations to the sampling context for sedimentological and geochronological analyses. For example, in Fig. 2 it would be nice to get a direct impression where the sampling site for OSL and radiocarbon dating is located and what the connection between the engravings and the geological record is. This link needs to be easily traceable in the main manuscript

as it provides a main line of evidence for the later interpretation e.g. presented in Fig. 6 and the discussion section.

- 4) Interpretation of playa sedimentation (e.g. line 360, or supplement 5 section 5.3.): the authors use playa sedimentation as a local paleoenvironmental records especially with regard to water availability, which generally is a good idea. However, the authors ignore in their analyses and interpretation of the record that wind erosion could have played a significant role in forming this playa sediment successions, which would imply that the claim of the first humid phase since the LGM is a bit on shaky ground. Alternatively, one could interpret that a previous humid phase is missing in the record either because there was not enough water available in the catchment to form playa sediments prior to this phase or the corresponding playa sediments had been subsequently eroded by wind erosion during a dryer phase. From my perspective are both scenarios possible with an almost equal probability.
- 5) Interpretation of the results (e.g. Fig. 6): as addressed already in my major comment#3 it is very difficult trace back the actual evidence that form the basis for the interpretations presented in this section and figure 6. As this is a scientific paper and journal, I think that the connection between the argument and the underlying scientific evidence should be as strong as possible. That said, I recommend to majorly revise Fig. 6 with the goal to make all links to the actual scientific evidence clearly visible. For example, the interpretation of the environmental context in the very right column, what is it actually based upon? The evidence needs to be directly traceable. Same with the columns human activity and rock art.
- 6) Supplement 5, Figure 42: the header “grainsize and texture” is a bit to vague. What statistical parameter did you use to describe the underlying distribution of grainsizes? Is it D50 or something else?
- 7) Supplement 5, Figure 42 and page 56, section 5.3: why was the sandy gravel layer only sampled for dating in JMI and not in ARN? With only one age on this unit the interpretation that it belongs to very late MIS5 (page 56, section 5.3 first para of this section) is shaky and I think this should be highlighted in the text.
- 8) Supplement 6: to show the general robustness of both the multiple-grain as well as the single-grain results the authors need to show dose recovery results, which is the standard performance test. Without dose recovery results the luminescence data is difficult to trust.
- 9) Supplement 6, section 6.1.5: to be able to judge on the appropriateness of the interpretation of the De distributions and the application of the different age models to obtain the burial dose I strongly recommend to provide De distributions (preferably as radial plots but KDE plots would also work) in supplement 6. Without De distribution is difficult to judge on the robustness of the obtained Db and thus OSL ages.
- 10) Supplement 6, section 6.1.5: The use of the FMM for sample JMI8-T1-4 seems problematic to me in this context.
 - a. To me the more appropriate age model for this sample would be the MAM especially when using the bootstrap version (Cunningham and Wallinga 2012) where sigma_b can be assigned with an uncertainty (e.g. $25 \pm 5\%$).
 - b. The FMM would only make sense if the sample is “contaminated” with young dose grains, something the authors recognize themselves (section 6.1.5, first

para, last sentences), as the MAM would have difficulties to handle these low-dose outliers/population. However, in the next paragraph of section 6.1.5 you write that sample JMI8-T1-4 showed high-dose grains presumably resulting from “contamination” from older and not younger grains. So, there is no added value in applying the FMM here.

- c. The way σ_b is determined by basically doing a sensitivity analysis using the FMM is not stringent if you ask me. The σ_b describes the unexplained scatter in the De distribution for a perfectly bleached and un-mixed sample. As you have samples that seem to behave this way, I strongly recommend to apply single-grain measurements to one of your well-behaving samples and use the obtained over-dispersion as input σ_b for Db determination of sample JMI8-T1-4. If you apply the bootstrap MAM (and I think you should) you can assign an uncertainty to this σ_b (e.g. 20 % relative uncertainty), which is more realistic than using σ_b without any uncertainty.

Some additional minor comments:

- 11) Line 196ff: Please rephrase these two sentences starting with “The engravers....”. There are non-scientific and speculative.
- 12) Line 204ff: also, here I recommend to tone down the overly sensational tone of this paragraph.
- 13) Figure 3: please provide the ages in the figure to make it easier to link engravings to the geochronological framework (also see my main comment#3). Also provide some more details on the kind of sand (e.g. with regard to texture, bedding structures) deposited here.
- 14) Line 400: “...pre-date the tradition of life-sized camel engravings...” → which is evidenced by?
- 15) Figure 7: where is North?
- 16) Supplement 6, page 59, section 6.1.4: please state size of the multiple-grain aliquots
- 17) Supplement 6, page 59, section 6.1.4: indicate in table 8 which ages are based on multiple-grain and which ages are based on single-grain measurements to make it easier to follow. That said, I’d also add the information on the age model applied to the table.
- 18) Supplement 6, page 59, section 6.1.4: why was an exponential plus linear fit of the dose response curve used? There is no reason based on physics to use this.

Best regards,
Tony Reimann
(University of Cologne)